# SIRT1 mediates obesity- and nutrient-dependent perturbation of pubertal timing by epigenetically controlling *Kiss1* expression

M.J. Vazquez[1,2,3,4], C.A. Toro [5], J.M. Castellano[1,2,3,4], F. Ruiz-Pino[1,2,3,4], J. Roa[1,2,3,4], D. Beiroa[6], V. Heras[1,2,3], I. Velasco[1,2,3,4], C. Dieguez[4,6], L. Pinilla[1,2,3,4], F. Gaytan[1,2], R. Nogueiras [4,6], M.A. Bosch[7], O.K. Rønnekleiv[7,8], A. Lomniczi [5], S.R. Ojeda[8] & M. Tena-Sempere [1,2,3,4,9]

Puberty is regulated by epigenetic mechanisms and is highly sensitive to metabolic and nutritional cues. However, the epigenetic pathways mediating the effects of nutrition and obesity on pubertal timing are unknown. Here, we identify Sirtuin 1 (SIRT1), a fuel-sensing deacetylase, as a molecule that restrains female puberty via epigenetic repression of the puberty-activating gene, *Kiss1*. SIRT1 is expressed in hypothalamic Kiss1 neurons and suppresses *Kiss1* expression. SIRT1 interacts with the Polycomb silencing complex to decrease *Kiss1* promoter activity. As puberty approaches, SIRT1 is evicted from the *Kiss1* promoter facilitating a repressive-to-permissive switch in chromatin landscape. Early-onset overnutrition accelerates these changes, enhances *Kiss1* expression and advances puberty. In contrast, undernutrition raises SIRT1 levels, protracts *Kiss1* repression and delays puberty. This delay is mimicked by central pharmacological activation of SIRT1 or SIRT1 overexpression, achieved via transgenesis or virogenetic targeting to the ARC. Our results identify SIRT1-mediated inhibition of *Kiss1* as key epigenetic mechanism by which nutritional cues and obesity influence mammalian puberty.

[1] Instituto Maimónides de Investigación Biomédica de Cordoba (IMIBIC), 14004 Cordoba, Spain. [2] Department of Cell Biology, Physiology and Immunology, University of Cordoba, 14004 Cordoba, Spain. [3] Hospital Universitario Reina Sofia, 14004 Cordoba, Spain. [4] CIBER Fisiopatología de la Obesidad y Nutrición, Instituto de Salud Carlos III, 14004 Cordoba, Spain. [5] Division of Genetics, Oregon National Primate Research Center/Oregon Health and Science University, Beaverton, OR 97006, USA. [6] Department of Physiology, Faculty of Medicine and CIMUS, University of Santiago de Compostela-Instituto de Investigación Sanitaria, 15782 Santiago de Compostela, Spain. [7] Department of Physiology and Pharmacology, Oregon Health and Science University, Portland, OR 97239, USA. [8] Division of Neuroscience, Oregon National Primate Research Center/Oregon Health and Science University, Beaverton, OR 97006, USA. [9] FiDiPro Program, University of Turku, Turku 20014, Finland. These authors contributed equally: M. J. Vazquez, C. A. Toro. Correspondence and requests for materials should be addressed to A.L. (email: lomniczi@ohsu.edu) or to M.T.-S. (email: fi1tesem@uco.es)

Sexual maturity and reproductive capacity are attained at puberty[1]. Understanding the intimate mechanisms controlling this process has gained new urgency in view of the observations that a secular trend towards an earlier puberty appear to be re-initiated[2,3]. This is worrisome because early puberty in humans is associated with a number of undesirable outcomes, such as cardiovascular disease, obesity, insulin resistance, hypertension, diabetes type 2, increased incidence of breast cancer, increased susceptibility to mental illness and behavioral disorders, and lower adult height[4–6].

Although puberty is under strong genetic determination[7], its timing is modulated by endogenous and exogenous cues[1]. Among those, nutritional and metabolic signals are known to play very prominent regulatory roles[8,9]. Consequently, conditions of metabolic stress, ranging from malnutrition to early-onset obesity, are linked to perturbed pubertal timing[10]; child obesity appears to be particularly relevant because of its association with early puberty worldwide[2,11]. Yet, despite the importance of these findings, the mechanisms conveying nutritional status information to the cellular networks controlling puberty remain ill defined.

The pubertal process depends on changes in the secretory activity of hypothalamic neurons producing gonadotropin-releasing hormone (GnRH). Because GnRH neurons themselves appear to be devoid of the appropriate sensing mechanisms[10], nutrition-dependent regulation must instead operate within cellular networks functionally connected to GnRH neurons. One such network is a group of neurons known as KNDy neurons because they produce Kisspeptin, NKB (neurokinin B) and Dynorphin[12–14]. They are located in the arcuate nucleus (ARC) of the hypothalamus[13,14], and drive the changes in GnRH secretion that set in motion the endocrine manifestations of puberty[15]. Because both *Kiss1* (encoding kisspeptin) and *Tac3* (encoding NKB) are essential for puberty to occur[16,17], KNDy neurons appear to be uniquely posed to serve as nodal portals for nutritional cues to influence reproductive development. In rodents, another population of neurons expressing *Kiss1*, but not *Tac3*, is located in the rostral hypothalamus, mainly at the anteroventral periventricular nucleus (AVPV)[18]. However, its role in the nutritional modulation of puberty has not been established.

Sirtuins are the homologs of the ageing-related factor, Silent Information Regulator 2 (Sir2), discovered in *Saccharomyces cerevisiae*[19–21]. Among sirtuins, SIRT1, with abundant expression in the brain and different metabolic peripheral tissues, is the most extensively studied[19]. SIRT1 operates as $NAD^+$-dependent deacetylase[22], acting on histones and other cellular targets to conduct a wide array of biological functions, including epigenetic control and modulation of life/health span[19–21]. Sirtuins act as bona fide cell energy sensors: activation of sirtuins is tightly coupled to changes in the availability of the metabolic cofactor, $NAD^+$, and the related intermediates, NADH and nicotinamide, so that conditions as caloric restriction and nutrient deprivation, which increase the $NAD^+/NADH$ or $NAD^+/$nicotinamide ratios, result in increased SIRT1 content and activation in various tissues[23,24]. CNS mapping studies in adult mice revealed abundant expression of *Sirt1* in key hypothalamic nuclei controlling neuro-vegetative functions, including the ARC; central SIRT1 activity seemingly plays a key role in controlling longevity[25], and might mediate the effects of caloric restriction on life/health span in mammals[20,24].

The nutritional status of an individual has strong environmental promptings, suggesting that nutrition may influence reproductive development via epigenetic mechanisms. Indeed, KNDy neurons are subjected to a repressive epigenetic control, imposed by the Polycomb (PcG) silencing complex that prevents the premature unleashing of the pubertal process[26]. This repressive influence is counter-balanced by the Trithorax group (TrxG), which operate as a central element of the activating epigenetic machinery that offsets PcG actions[27]. Yet, little is known about putative epigenetic regulatory mechanisms conveying nutritional information to KNDy neurons. Because SIRT1 —as energy-sensing molecule that allows the cell to respond to both reduction and increases in nutrient availability[19,24]—is activated in the adult hypothalamus by decreased nutrient availability[28] and is abundant in the ARC[28–30], we explored the notion that SIRT1 is a key component of the epigenetic machinery that regulates the timing of puberty by conveying nutritional information to KNDy neurons. We also evaluated the potential pathophysiological role that SIRT1 may play in eliciting pubertal perturbations associated with early-onset obesity and undernutrition.

Our present results support an essential role of SIRT1 as a central hub of an epigenetic mechanism mediating the effect of nutritional cues on female puberty, through regulation of *Kiss1* transcription. Our findings show that SIRT1 is expressed in hypothalamic Kiss1 neurons and suppresses *Kiss1* expression, via interaction with the Polycomb silencing complex. As puberty approaches, SIRT1 is evicted from the *Kiss1* promoter facilitating a switch in local chromatin configuration from repressive to permissive. As evidence for their translational relevance, these changes are accelerated by early-onset obesity, which induces precocious puberty, and postponed by undernutrition, which delays puberty. The pivotal role of SIRT1 on the latter phenomenon is evidenced by the ability of central pharmacological activation of SIRT1 or SIRT1 overexpression, either globally or targeted to the ARC, to mimic the pubertal delay evoked by undernutrition.

## Results

**Developmental and nutritional changes in hypothalamic SIRT1 during puberty.** SIRT1 content decreased in the preoptic area (POA)-medial basal hypothalamus (MBH) unit analyzed in toto before and during the initiation of female puberty. This change, already apparent by PND10 (Supplementary Figure 1), became firmly established by the end of juvenile development (PND28), and reached prominence between the end of juvenile development (PND28) and the peripubertal period (PND 36; Fig. 1a). The decrease was also apparent when the MBH and POA were analyzed separately, although it was proportionally greater in the MBH (>65%) than in the POA (Fig. 1b). The drop in hypothalamic SIRT1 content that occurs between PND10 and PND36 coincided with a significant increase in *Kiss1* expression levels, both in the POA and MBH (Fig. 1c). *Tac3* mRNA content in the MBH also increased during this period (Supplementary Figure 2a).

The relationship that exists between nutritional status and hypothalamic SIRT1 was evidenced by a reduction in SIRT1 abundance observed at PND29 in overnourished (ON) animals (Fig. 1d). This change was associated with a significant elevation in both *Kiss1* and *Tac3* mRNA levels (Fig. 1e and Supplementary Figure 2b). ON rats also displayed increased body weight (Fig. 1f) and early puberty, as assessed by the age at vaginal opening (28.1 ± 0.4 d vs. normally fed (NN) controls, 32 ± 0.3 d, $p < 0.01$; Student's $t$ test) and at first ovulation, two well-established signs of puberty. Thus, while >80% of obese animals exhibited vaginal opening (Fig. 1g) and corpora lutea (CL) (Fig. 1h) at PND29 ($p < 0.001$, $X^2$ test), none of the normally fed controls had signs of puberty at this age (Fig. 1g, h). Both the uterus (index of estrogen stimulation; Fig. 1i) and serum LH levels (Fig. 1j; reflecting hypothalamic-pituitary activity) were significantly greater in obese than in normally fed rats. These results show that

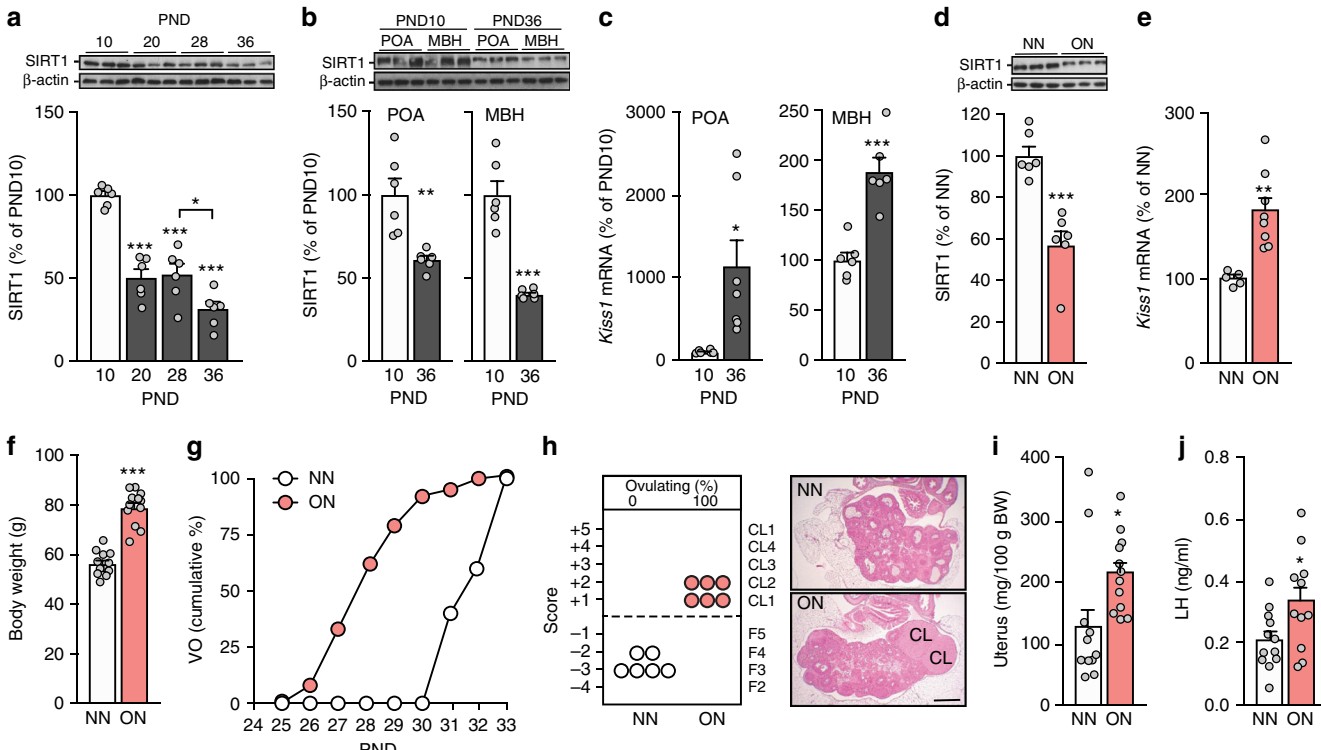

**Fig. 1** Changes in hypothalamic SIRT1 content and pubertal timing induced by overnutrition. **a** SIRT1 content in the medial basal hypothalamus (MBH)/ Preoptic area (POA) of female rats at the infantile (PND10; white bars), early juvenile (PND20; gray bars), late juvenile/pre-pubertal (PND28; gray bars), and peripubertal (PND36; gray bars) stages of reproductive development, determined by western blot analysis. *$p < 0.05$ vs. PND28; ***$p < 0.001$ vs. PND10 (one-way ANOVA followed by post-hoc Tukey test). **b**, **c** SIRT1 protein and *Kiss1* mRNA content in the MBH and POA at PND10 and PND36; for data in **a**−**c**, $n = 6$ animals per group. **d**, **e** SIRT1 content and *Kiss1* mRNA levels in the MBH of late juvenile, 29-day-old female rats fed normally (NN: normal nutrition; white bars) or subjected to nutritional excess (overnutrition, ON; light red bars) during postnatal development. **f** Body weight of NN and ON animals. **g** Cumulative percent of NN and ON animals at vaginal opening (VO). **h** Histological score of follicular development/ovulation and representative images of ovarian maturation in NN and ON groups; CL: corpus luteum. **i** Uterine weight in NN and ON animals. **j** Plasma LH in the same animals. The bar histograms represent the mean ± SEM. In panels **b**-**f**, **i** and **j**, *$p < 0.05$; **$p < 0.01$; ***$p < 0.001$ (two-sided Student's *t* test). For protein analyses in panels **a**, **b** and **d**, three representative bands per group, run in the same original western blots, are presented. The scale bar in panel **h** corresponds to 600 µm. Total group sizes were: NN = 11 and ON = 12; while phenotypic and hormonal parameters were assayed in the whole groups, hypothalamic protein/RNA (**d**, **e**) and ovarian histological (**h**) analyses were conducted in a representative subset of randomly assigned samples from each group, with the following distribution: **d**, **e** NN = 5; ON = 6−8; **h** $n = 6$ in each group

overnutrition instituted during early postnatal life reduces hypothalamic SIRT1 content and causes precocious puberty.

In contrast, undernourishment (UN) imposed at the beginning of the juvenile period resulted in increased hypothalamic SIRT1 content and decreased *Kiss1* expression, assessed at PND36 (Fig. 2a, b); a non-ignificant decrease in *Tac3* levels was also observed (Supplementary Figure 2c). UN also caused a reduction in body weight (Fig. 2c), delayed vaginal opening (Fig. 2d), and ovulatory failure (Fig. 2e). Both uterine weight (Fig. 2f) and LH levels (Fig. 2g) were reduced in underfed animals. Thus, prepubertal undernutrition increases SIRT1 content and lowers *Kiss1* expression in the hypothalamus, and delays female puberty.

**Functional manipulation of hypothalamic SIRT1 alters puberty.** To determine if the inverse correlation that exists between hypothalamic SIRT1 and the time of female puberty involves a causal relationship, we employed three different approaches. First, we studied transgenic mice overexpressing SIRT1[31,32] (SIRT1-Tg). Second, we used the allosteric SIRT1-activator, SA3, to stimulate centrally endogenous SIRT1 activity[33]. Third, we used a virogenetic approach to selectively increase SIRT1 expression in the ARC of the hypothalamus. SIRT1-Tg mice had normal body

weight (Fig. 3a) and displayed elevated SIRT1 content in the hypothalamus (Fig. 3b), coupled to a decrease in the abundance of histone 3 acetylated at lysines 9/14 (H3K9/14Ac) (Fig. 3c), likely reflecting an increase in SIRT1-dependent deacetylase activity. In addition, SIRT1-Tg mice showed a significant reduction in hypothalamic *Kiss1* mRNA levels (Fig. 3d), but not *Tac3* mRNA (Supplementary Figure 2d). Importantly, SIRT1-Tg animals had a strikingly delayed vaginal opening (32.7 ± 1.3 d vs. WT, 28.3 ± 0.7 d, $p = 0.008$; Student's *t* test). By PND33, all WT mice had vaginal opening, in contrast to only 35% of the SIRT-Tg mice ($p < 0.0001$; $X^2$ test) (Fig. 3e); none of the Tg mice had ovulated at this time (Fig. 3f). Similar to undernourished rats, SIRT1-Tg animals had reduced LH levels at puberty (Fig. 3g). Thus, as initially suggested by others[34], our data conclusively document that transgenic overexpression of SIRT1 delays reproductive maturation.

We next injected SA3 into the lateral ventricle of immature female rats daily between PND26 and PND35. Like SIRT1-Tg mice, SA3-treated rats had a normal body weight (Fig. 4a), but delayed vaginal opening. At PND33, only 20% of SA3-injected animals had vaginal opening in contrast to 75% of vehicle-injected rats ($p = 0.06$; $X^2$ test; Fig. 4b). At the time of tissue collection (PND35), 40% of SA3-treated rats had not ovulated

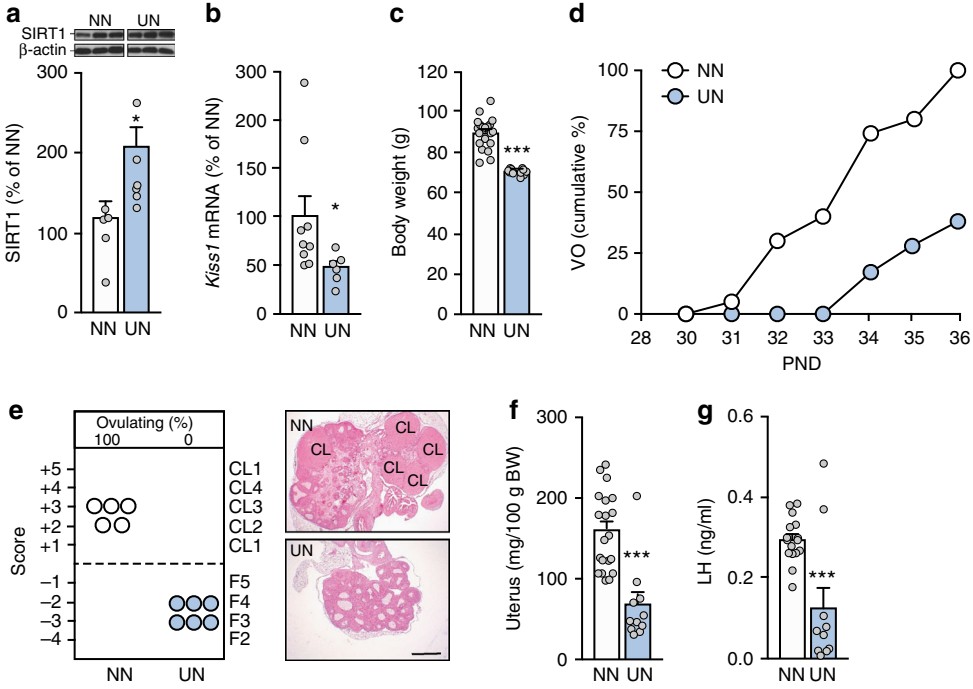

**Fig. 2** Changes in hypothalamic SIRT1 and pubertal timing induced by undernutrition. **a**, **b** SIRT1 content and *Kiss1* mRNA levels at PND36 in the MBH of animals fed a normal diet (NN; white bars) or given restricted access to food (undernutrition, UN; light blue bars) during prepubertal development. **c** Body weight of NN and UN animals. **d** Cumulative percent of NN and UN animals showing vaginal opening (VO). **e** Histological score of follicular development/ ovulation and representative images of ovarian maturation in the NN and UN groups; CL: corpus luteum. **f** Uterine weight and **g** Plasma LH levels. The bar histograms represent the mean ± SEM. *$p < 0.05$; ***$p < 0.001$ vs. NN group (two-sided Student's $t$ test). For protein analyses in panel **a**, three representative bands per group, run in the same original western blots, are presented. The scale bar in panel **e** corresponds to 600 μm. Total group sizes were: NN = 20 and ON = 10; while phenotypic and hormonal parameters were assayed in the whole groups (in the case of LH levels, for all serum samples that were available), hypothalamic protein (**a**) and RNA (**b**), as well as ovarian histological (**e**) analyses were conducted in a representative subset of randomly assigned samples from each group, with the following distribution: **a** n = 5; **b** n = 6−8; **e** n = 5−6 determinations

(Fig. 4c). Mimicking the phenotype of SIRT1-Tg mice, SA3-injected rats had lighter uteri (Fig. 4d) and reduced serum LH levels (Fig. 4e). In addition, hypothalamic SIRT1 content was increased (Fig. 4f), and the abundance of H3K9/14Ac was reduced (Fig. 4g), likely due to increased SIRT1-dependent deacetylase activity. Noteworthy, SA3-treated rats had reduced *Kiss1* expression in the MBH (Fig. 4h), but unaltered *Tac3* levels (Supplementary Figure 2e).

Given the location of KNDy neurons in the ARC, we used a virogenetic approach to target SIRT1 overexpression to this hypothalamic region. Stereotaxic-guided delivery of an AAV vector expressing SIRT1 resulted in increased SIRT1 content (Fig. 5e), and decreased ratios of acetylated K9/14 H3 vs. total H3 (AcH3/H3) in the MBH (Fig. 5e). Congruent with the results observed using SIRT1-Tg mice and rats centrally injected with SA3, increasing SIRT1 abundance in the ARC resulted in delayed vaginal opening; while >90% of rats injected with control AAV had vaginal opening by PND33, <40% of AAV-SIRT1 injected rats displayed VO at this age ($p < 0.01$; $X^2$ test; Fig. 5f). Likewise, AAV-SIRT1 rats had a markedly delayed first estrus (Fig. 5g) and stunted follicular maturation, with 60% of the animals failing to ovulate at PND36 (Fig. 5h). Pubertal female rats injected with AAV-SIRT1 displayed also a significant decrease in ovarian weight (Fig. 5i). Thus, by selectively enhancing SIRT1 abundance in the ARC (and hence, preventing the drop of expression that occurs before puberty), female reproductive maturation was delayed.

**SIRT1 is expressed in Kiss1 neurons where it is modulated by nutritional cues.** Double fluorescence in situ hybridization

(FISH) and single cell (sc)-PCR demonstrated that KNDy neurons in the MBH/ARC do express *Sirt1* mRNA (Fig. 6a, b). Detailed quantification demonstrated that ~50% of all Kiss1-expressing neurons in the ARC express *Sirt1* (Fig. 6b). We confirmed this colocalization using a double immunohisto-fluorescence approach. In fact, in agreement with our previous data (see Figs. 1 and 2), SIRT1 content in Kiss1 neurons was decreased in pubertal overfed ON rats (Fig. 6c), while it was increased in undernourished UN rats (Fig. 6d). Altogether, these data support the concept that prepubertal nutrient manipulation can affect KNDy neuron activity (as evidenced by changes in *Kiss1* expression) via changes in SIRT1 that occur within KNDy neurons themselves.

**SIRT1 evokes a repressive histone configuration at *Kiss1* promoter that abates at puberty.** Undernourishment and obesity/ overnutrition affected hypothalamic *Kiss1* expression at puberty in opposite directions: the former decreasing *Kiss1* mRNA levels, the latter increasing gene expression. To determine if these effects are directly exerted on *Kiss1* neurons, we interrogated the *Kiss1* promoter, using ChIP−qPCR assays, targeting a promoter region involved in the epigenetic control of *Kiss1* expression[26]. We found that SIRT1 is evicted from the *Kiss1* promoter at the completion of puberty (PND36) (Fig. 7a). This loss takes place earlier (at PND29) in overfed animals, but fails to occur in undernourished rats (Fig. 7a). Two activating histone modifications (H3K9ac and H4K16ac), normally removed by SIRT1-mediated deacetylation, showed a pattern of abundance at the *Kiss1* promoter opposite to that of SIRT1. While the content of both marks increases during the pubertal transition, this increase

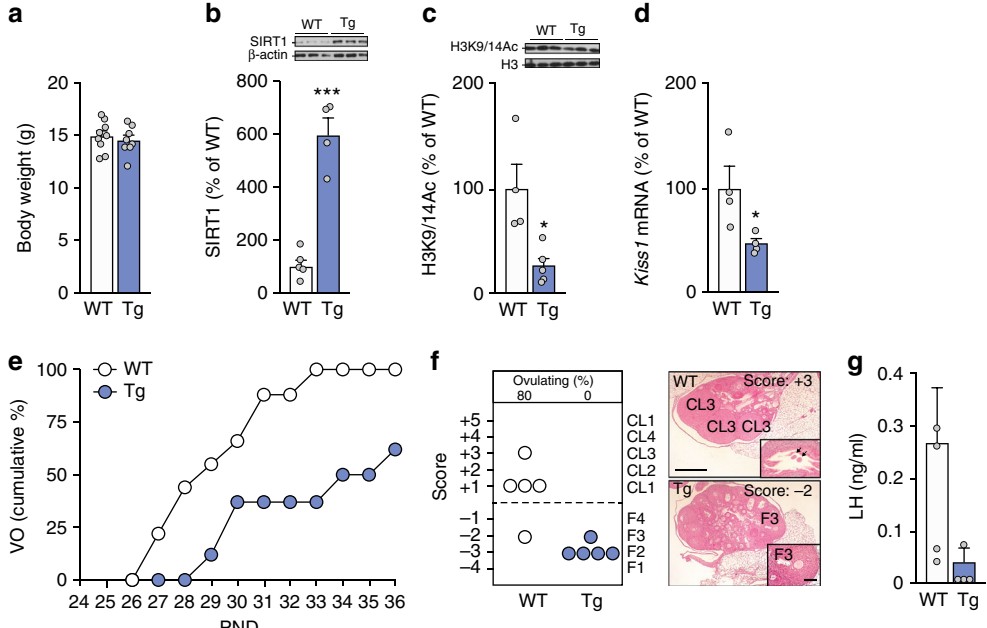

**Fig. 3** Transgenic SIRT1 excess in female mice delays pubertal timing. SIRT1 content and indices of somatic/pubertal maturation in a transgenic model of moderate *Sirt1* overexpression (Tg; cyan bars) and wild-type (WT; white bars) controls measured at PND32 are shown. **a** Body weight. **b** Hypothalamic SIRT1 content. **c** Hypothalamic H3K9/14Ac content. **d** Hypothalamic *Kiss1* mRNA levels. **e** Cumulative percent of animals showing vaginal opening (VO). **f** Histological score of follicular development/ovulation and representative images of ovarian maturation; CL: corpus luteum, F3: stage-3 follicle; released oocytes are denoted by arrows in the inset. **g** Plasma LH levels. The bar histograms represent the mean ± SEM. *$p < 0.05$; ***$p < 0.001$ (two-sided Student's *t* test). For protein analyses in panels **b** and **c**, three representative bands per group, run in the same original western blots, are presented. The scale bars correspond to 600 μm in panel **f**, and 200 μm in the inset. Total group sizes were: WT = 10 and TG = 8; while phenotypic parameters (BW) were assayed in the whole groups, molecular and hormonal analyses were conducted in a representative subset of randomly assigned samples from each group ($n = 5$), which were euthanized at the time of puberty. Note that due to a technical loss in serum samples, determinations in **e** were $n = 4$ per group

occurs prematurely in overfed animals and is averted in undernourished rats (Fig. 7a). In contrast, the content of H3K27me3, a repressive histone modification catalyzed by the PcG complex[35], paralleled that of SIRT1 at the *Kiss1* promoter, decreasing at the completion of normal puberty, diminishing earlier in obese rats, and increasing at the promoter of underfed rats (Fig. 7a).

Because the Trithorax group (TrxG) of proteins is the main epigenetic force counterbalancing the repressive actions of the PcG complex[27,36], we determined if the abundance of H3K4me3, an activating histone mark catalyzed by TrxG[27,36], changes at *Kiss1* promoter in response to nutritional manipulations. While H3K4me3 content, measured at the end of juvenile development (PND29), was greatly increased in obese rats, undernutrition did not significantly alter H3K4me3 levels at either PND29 or PND36 (Fig. 7a). This suggests that TrxG influence might be more prominent in conditions of energy surplus. The genomic specificity of the changes described above was documented by ChIP assays targeting intron 2 of *Kiss1*. As shown in Fig. 7b, none of the changes in histone mark abundance observed at the *Kiss1* promoter occurred at intron 2 of this gene.

Although pubertal expression of *Tac3* was only mildly influenced by the nutritional status, we explored SIRT1-related interactions with the *Tac3* promoter, a genomic region recently shown to be subjected to epigenetic control at puberty[27]. SIRT1 association to the *Tac3* promoter decreased between late juvenile development and the completion of puberty, but neither obesity nor undernourishment altered this association (Supplementary Figure 3a). Similarly, nutritional manipulations did not change the H3K9ac and H4K16ac content at the *Tac3* promoter on PND29, but underfeeding reduced the content of H3K9ac mark on PND36 (Supplementary Figure 3a). The abundance of the

repressive histone mark, H3K27me3, at the *Tac3* promoter increased mildly at PND36 in underfed rats, without any changes in the content of H3K4me3 (Supplementary Figure 3a). Finally, neither normal puberty nor nutritional manipulations altered the association of SIRT1 to, or the histone configuration at, the *Pdyn* promoter (Supplementary Figure 3b). Thus, it appears that SIRT1 prevents the premature activation of KNDy neurons in immature animals by mostly repressing *Kiss1* activity, and that earlier or late puberty in over- and underfed animals, respectively, correlate with changes in SIRT1 binding and activity mainly at the *Kiss1* promoter. It is also clear that these modifications do not occur in the POA, which contains the AVPV; the only changes in this region were a tendency for a decrease of SIRT1 association to the *Kiss1* promoter and an increase in the content of the activating histone mark, H4K16ac, observed in overfed ON rats (Supplementary Figure 4).

**SIRT1 cooperates with EED to repress *Kiss1* promoter activity.** The parallel loss of SIRT1 and H3K27me3 association to the *Kiss1* promoter elicited by pubertal maturation and overnutrition suggests that SIRT1 and the PcG complex act in a coordinated fashion to repress *Kiss1* activity in KNDy neurons of prepubertal animals. Promoter assays revealed that *Kiss1* promoter activity is modestly reduced by either SIRT1 or EED, a protein required for PcG silencing activity[37], but is more clearly inhibited by the simultaneous presence of both proteins (Fig. 8a). Coimmuno-precipitation studies demonstrated that SIRT1 and EED interact physically (Fig. 8b). When SIRT1 was overexpressed in a rat (R22) immortalized hypothalamic cell line (Fig. 8c), there was increased interaction of SIRT1 protein to the *Kiss1* promoter

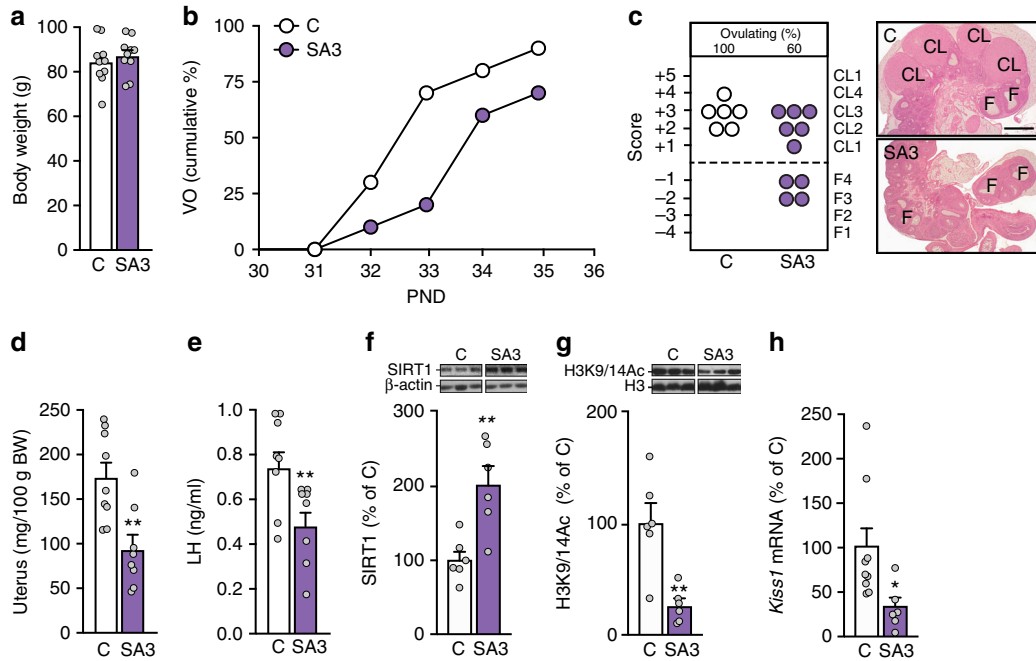

**Fig. 4** Central pharmacological activation of SIRT1 delays pubertal timing. Indices of somatic/pubertal maturation in immature female rats following central, pharmacological SIRT1 activation by the allosteric SIRT1-activator, SA3, are shown. **a** Body weight of controls (C; white bars) and SA3-treated (bars in violet) rats. **b** Cumulative percent of C and SA3-treated rats showing vaginal opening (VO). **c** Histological score of follicular development/ovulation and representative images of ovarian maturation; CL: corpus luteum, F: follicle. **d** Uterine weight. **e** Serum LH levels. **f** Hypothalamic SIRT1 content. **g** Hypothalamic H3K9/14Ac content. **h** *Kiss1* mRNA levels. The bar histograms represent the mean ± SEM. *$p < 0.05$; **$p < 0.01$ (two-sided Student's *t* test). For protein analyses in panels **f** and **g**, three representative bands per group, run in the same original western blots are presented. The scale bar in panel **c** corresponds to 600 μm. Total group sizes were: C = 10 and SA3 = 10; while phenotypic and hormonal parameters were assayed in the whole groups (in the case of LH levels, for all serum samples that were available), hypothalamic protein (**f**, **g**) and RNA (**h**) analyses were conducted in a representative subset of randomly assigned samples from each group, with the following distribution: C: $n = 6-8$ (protein and RNA, respectively); SA3: $n = 6$

(Fig. 8d) and a significant decline in *Kiss1* mRNA content (Fig. 8e). This was accompanied by decreased content of the activating mark H3K9ac and the TrxG-dependent mark H3K4me3, and increased abundance of the PcG-catalyzed mark H3K27me3 (Fig. 8f). Concomitantly, EED recruitment to the promoter rose (Fig. 8g). These results indicate that SIRT1 and EED act together at the *Kiss1* promoter in KNDy neurons to turn a permissive histone landscape into a repressive histone configuration, resulting in decreased *Kiss1* transcription.

## Discussion

Our understanding of the molecular mechanisms responsible for the precise timing of puberty has expanded recently, due in part to the recognition of the role of epigenetic mechanisms in the control of normal pubertal maturation[26,38]. However, characterization of the repertoire of epigenetic regulatory molecules, especially as it pertains to the nutritional gating of puberty and its deregulation in conditions of metabolic stress, remains incomplete. In principle, molecules that use epigenetic mechanisms to link metabolic activity with neuroendocrine reproductive control should have four basic attributes that enable them to perform these functions: (a) to be able to engage directly with the basic machinery of epigenetic regulation; (b) to respond to alterations in cellular metabolism with meaningful changes in expression and/or activity; (c) to sense nutrient-dependent changes in the activity of nodal components of relevant metabolic pathways; and (d) to modify the activity of genes directly involved in the control of GnRH release. Our results conclusively identify SIRT1, the mammalian ortholog of Sir2[19,24], as a regulatory molecule that

meets all of these criteria, and unveil a central regulatory hub, involving SIRT1-mediated transcriptional repression of *Kiss1* in KNDy neurons of the ARC, as an essential mechanism regulating the timing and metabolic gating of puberty.

The inhibitory influence of central SIRT1 signaling on pubertal timing was disclosed by a combination of physiological and molecular approaches applied to various preclinical models. Not only the hypothalamic content of SIRT1 declines during postnatal/pubertal maturation, but translational models of precocious and delayed puberty, caused by early over- or undernutrition of female rats, display coherent changes in hypothalamic SIRT1 levels: advanced puberty is accompanied by decreased hypothalamic SIRT1 content, whereas delayed puberty associates with increased SIRT1 levels. The causal nature of these associations is demonstrated by our studies involving functional manipulation of SIRT1 levels/activity. Thus, SIRT1 overexpression by either transgenesis, using a validated mouse model[31,32], or directed to the ARC via an AAV-mediated, stereotaxic-guided targeting approach, caused a delay in pubertal progression, reminiscent of that caused by undernutrition. A similar effect occurred following the pharmacological enhancement of SIRT1 activity in the brain. The changes in hypothalamic SIRT1 abundance caused by these experimental manipulations were related to inverse changes in *Kiss1* expression in vivo, suggesting a key role for *Kiss1* as putative conduit for the regulatory actions of SIRT1 on pubertal timing.

KNDy neurons in the ARC are a well-conserved, indispensable component for the control of pulsatile secretion of GnRH and gonadotropins[39], and therefore the timing of puberty in mammals. In line with this key role, our expression and functional

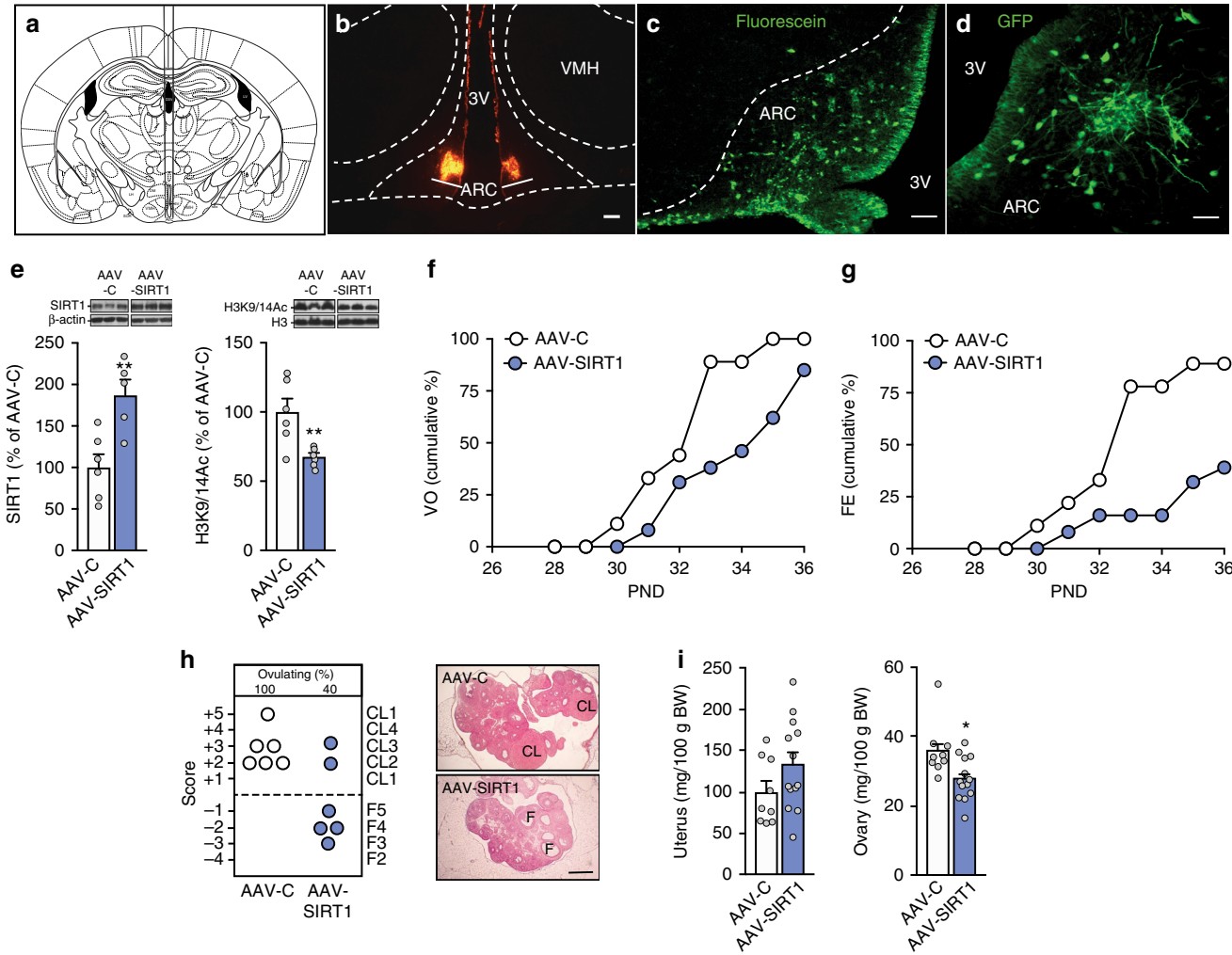

**Fig. 5** Virogenetic overexpression of SIRT1 in the ARC delays pubertal timing. **a** Scheme illustrating the stereotaxic delivery of AAV to the ARC. **b** Representative images of bilateral targeting of the ARC, denoted by fluorescence latex microspheres (beads). **c** Detail of neuronal targeting in the ARC by stereotaxic injection, denoted by fluorescein. **d** AAV-mediated infection of ARC neurons by stereotaxic delivery, denoted by GFP labeling. **e** SIRT1 content and acetylated K9/14 H3 vs. total H3 (AcH3/H3) ratios in the MBH of immature rats injected with AAV overexpressing SIRT1 (AAV-SIRT1) or their respective controls (AAV-C). **f, g** Cumulative percent of AAV-C- and AAV-SIRT1-treated rats showing vaginal opening (VO) or first estrus (FE). **h** Histological score of follicular development/ovulation and representative images of ovarian maturation; CL: corpus luteum, F: follicle. **i** Uterine and ovarian weights in AAV-C and AAV-SIRT1 groups. The bar histograms represent the mean ± SEM. *$p < 0.05$; **$p < 0.01$ (two-sided Student's $t$ test). For protein analyses in panel **e**, three representative bands per group, run in the same original western blots, are presented. The scale bars correspond to 100 µm in panels **b**, **c**, 200 µm in panel **d**, and 400 µm in panel **h**. Total group sizes were: AAV-C = 9 and AAV-SIRT1 = 13; while phenotypic parameters were assayed in the whole groups, hypothalamic protein (**e**) and ovarian histological (**h**) analyses were conducted in a subset ($n = 6$) of randomly assigned samples from each group. Bar graphs of the AAV-C group are in white, while bars from AAV-SIRT1 are in blue

analyses demonstrate that SIRT1 targets this neuronal population to metabolically gate pubertal timing, mainly by repressing *Kiss1* transcription. This contention is supported by the demonstration that (a) SIRT1 (mRNA and protein) is expressed in KNDy neurons, with a steady-state average of >50% of KNDy neurons expressing *Sirt1*; (b) *Kiss1* expression is suppressed in the MBH of pubertal female rats after transgenic overexpression and pharmacological activation of SIRT1 in vivo; (c) SIRT1 reduces *Kiss1* expression in vitro; and (d) SIRT1 content in ARC KNDy neurons and SIRT1 recruitment to the *Kiss1* promoter in vivo change according to the nutritional status of the animal; while overnutrition decreases neuronal content and promoter association, undernutrition increases both parameters. In addition, (e) virogenetic overexpression of SIRT1 in the ARC delays puberty onset. Although the potential contribution of SIRT1 to regulating the activity of other hypothalamic cell types remains a distinct

possibility, our findings strongly suggest that SIRT1 is a bona fide energy sensor that operates in KNDy neurons to transduce metabolic information into changes of *Kiss1* transcriptional activity. Importantly, proper progression through normal puberty appears to also require the timed eviction of SIRT1 from the *Kiss1* promoter, as a permissive requisite for the upsurge of hypothalamic *Kiss1* expression that occurs during the pubertal transition[18].

The molecular underpinnings of this phenomenon have been also disclosed by our work. Eviction of SIRT1 from the *Kiss1* promoter during puberty is associated with increased abundance of activating histone modifications (H3K9ac, H4K16ac, H3K4me3), and a decrease in the repressive histone mark H3K27me3 catalyzed by the PcG complex[35]. These changes appear to occur selectively in ARC KNDy neurons, since similar modifications were not detected in the AVPV region, which

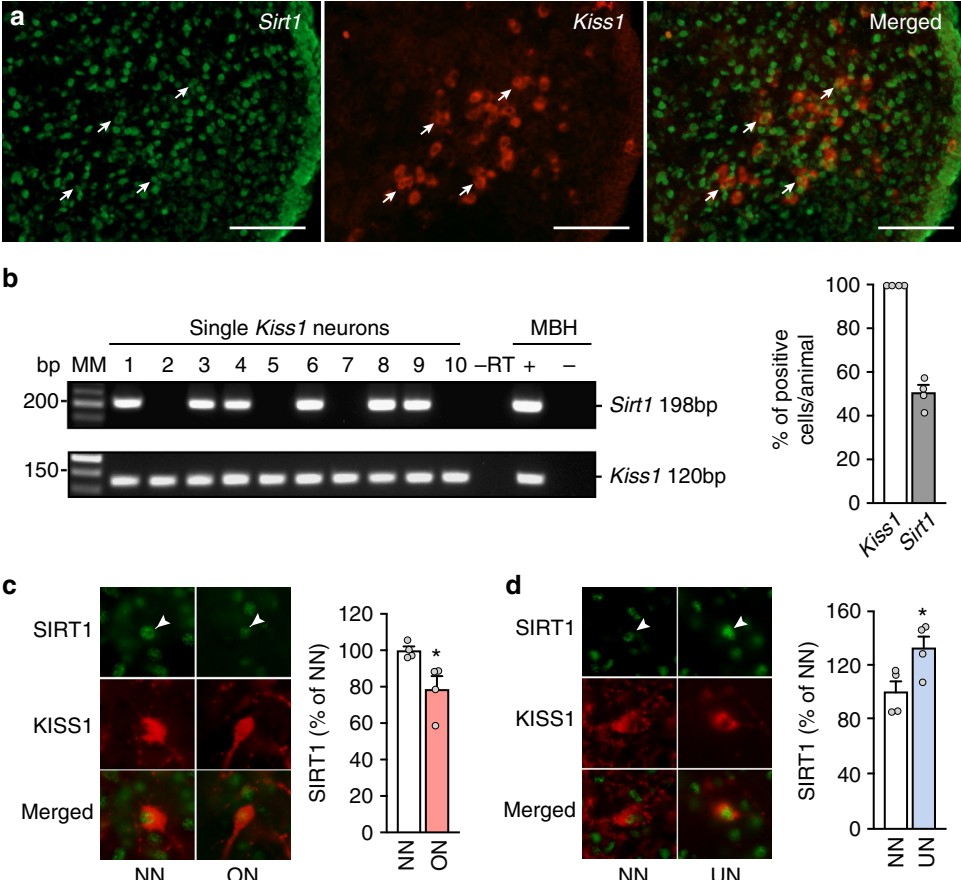

**Fig. 6** Sirt1 is expressed in KNDy neurons and SIRT1 content changes with nutritional status. **a** Fluorescent in situ hybridization (FISH) using a cRNA complementary to *Kiss1* mRNA (red) and a cRNA probe recognizing *Sirt1* mRNA (green) in KNDy neurons of the ARC. Arrows point to examples of double labeled cells. Scale bars, 100 μm. **b** A representative gel illustrating the presence of *Sirt1* and *Kiss1* mRNA in ten eGFP-tagged mouse KNDy neurons (1–10) of adult ovariectomized female mice as detected by single-cell (sc) PCR. As a negative control, the PCR reaction was performed on a Kiss1-GFP cell in the absence of reverse transcription (−RT). Other negative controls (showing no PCR product) included harvested aCSF and water blank. RNA extracted from the MBH was included as a positive control (+, with RT) and negative control (−, without RT). MM molecular markers, bp base pairs. The bar graph represents the mean ± SEM of the percentage of KNDy neurons expressing *Sirt1* per animal from four mice and a total of 96 harvested *Kiss1*-positive cells. **c**, **d** Immunohistofluorescence detection of SIRT1 (green) and kisspeptin (red) from the ARC of NN (normal-nutrition) vs. ON (overnutrition) rats at PND29 (panel **c**), or NN vs. UN (undernutrition) animals at PND36 (panel **d**). Bar graphs represent the mean ± SEM of the percentage of SIRT1 signal intensity in kisspeptin-positive cells from 30 to 50 neurons per animal, normalized against NN data. *$p < 0.05$ (two-sided Student's $t$ test) ($n = 4$ animals per group). NN = white bars; ON = light red bars; UN = light blue bars

contains a population of Kiss1 neurons involved in the pre-ovulatory surge of gonadotropins[18]. Moreover, our in vitro analyses demonstrate that SIRT1 recruits the PcG member, EED, to the *Kiss1* promoter, where it enhances the repressive action of SIRT1 on *Kiss1* transcription. Therefore, SIRT1 appears to operate as central epigenetic link between energy status and *Kiss1* expression in KNDy neurons, by changing the histone landscape of the *Kiss1* promoter. This molecular switch from a repressive to a permissive histone configuration is seemingly dictated by the nutritional influence on SIRT1 activity at the *Kiss1* promoter: conditions of energy excess, which induce precocious puberty, accelerate the removal of the epigenetic brake imposed by SIRT1/PcG on *Kiss1* expression. In contrast, persistent energy deficit, which causes pubertal delay, prolongs this repressive influence (Fig. 9). Our results pave the way for the identification of upstream regulators of hypothalamic SIRT1 that may, thereby, modulate pubertal timing. Besides nutritional cues, these might include metabolic hormones with known roles in puberty control, such as ghrelin, which has been shown to increase hypothalamic SIRT1 activity to regulate energy homeostasis in adulthood[40], and to delay puberty onset[41]. Regulation of SIRT1 in KNDy neurons

by ghrelin and/or other metabolic transmitters (e.g., adipokines), which are likely deregulated in some of our preclinical models of pubertal perturbation (e.g., early over- and undernutrition), as well as the almost certain contribution of epigenetic factors, warrants future investigation.

In the last decades, SIRT1 has been recognized as master molecular link (and potential druggable target) for the integration of metabolic sensing and key developmental and neuroendocrine phenomena[19–21]. Interestingly, genetic loss-of-function studies in mice had previously suggested a developmental role of SIRT1 in early stages of GnRH neuron maturation; mice with congenital *Sirt1* deficiency display hypogonadotropic hypogonadism (HH) due to defective neuronal migration[42,43]. Our work unveils another dimension of SIRT1 biology in the hypothalamic control of neuroendocrine function. Hitherto known to be an essential factor for the control of longevity and healthy ageing[21,44], SIRT1 appears to also function as fundamental epigenetic conduit that links obesity and nutritional status with changes in pubertal timing. This might not be the only puberty-related function of SIRT1 in the developing brain. Because the *SIRT1* locus in humans is strongly associated with susceptibility to major

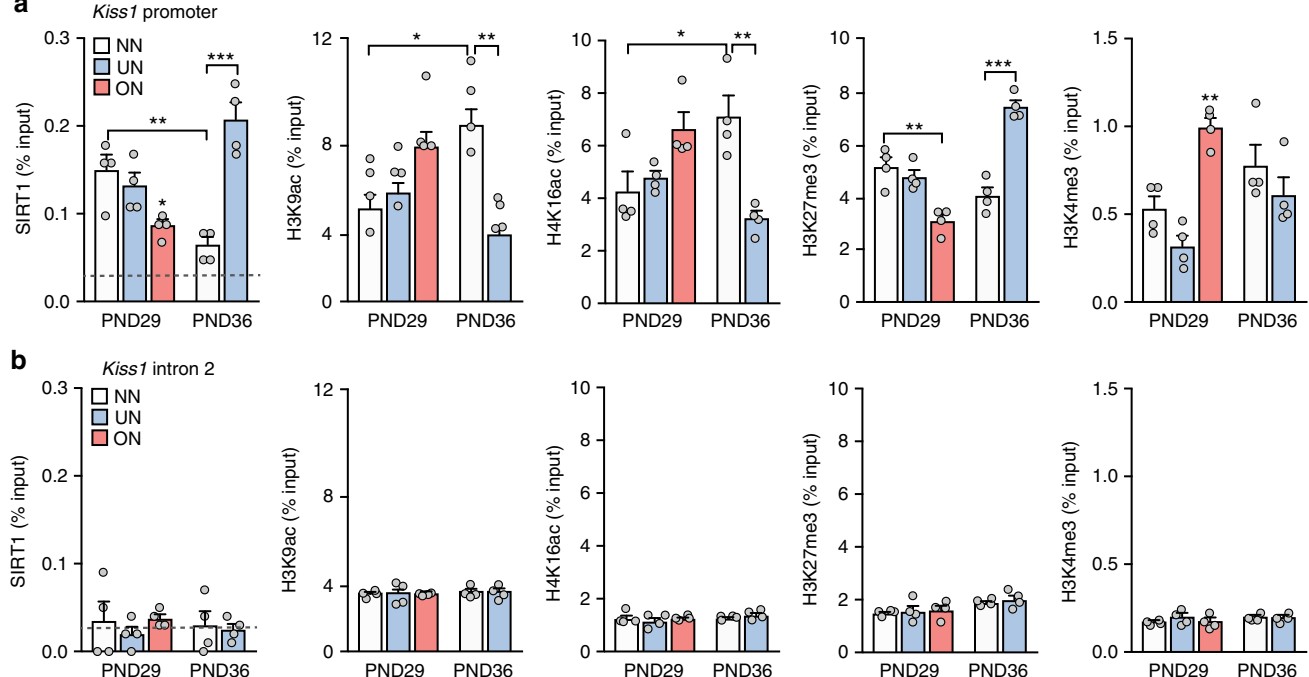

**Fig. 7** Sirt1 regulates the chromatin architecture at the Kiss1 promoter. **a** Chromatin immunoprecipitation (ChIP) assays showing the association of SIRT1, H3K9Ac, H4K16Ac, H3K27me3 and H3K4me3 to the *Kiss1* promoter in the MBH of female rats fed normally (NN) or subjected to postnatal nutritional manipulation (ON: overnutrition, UN: undernutrition). **b** Lack of changes in SIRT1, H3K9Ac, H4K16Ac, H3K27me3, and H3K4me3 association to intron 2 of the *Kiss1* gene. The results are expressed as % of the signal generated by input DNA. The bar histograms represent the mean ± SEM. *$p < 0.05$; **$p < 0.01$; ***$p < 0.001$ (one-way ANOVA followed by the Student−Newman−Keuls). For all panels, $n = 4$ animals/group. NN = white bars; ON = light red bars; UN = light blue bars

depression[45], and an increase in SIRT1 abundance in certain brain areas, such as the nucleus Accumbens, induces depressive-like behaviors in mice[46], failure of brain SIRT1 expression to decrease in a timely fashion during the adolescence transition to adulthood might contribute to the well-known exacerbation of depressive disorders at puberty[47], some of which occur in the context of obesity[48].

## Methods

**Animals**. We used Wistar female rats bred in the vivarium of the University of Cordoba and housed under constant conditions of light (14/10 h light/dark cycle) and temperature (22 ± 1 °C), with free access to tap water and standard laboratory chow, unless otherwise stated (see models of nutritional stress). For experiments involving functional manipulations, either nutritional, pharmacologic or virogenetic, the animals were randomly assigned to the different experimental groups. We also used transgenic female mice overexpressing *Sirt1* (Sirt1-Tg; kindly provided by Dr. Manuel Serrano, Institute for Research in Biomedicine (IRB), Barcelona, Spain). These animals were generated as previously described[31,32]; they were bred in the vivarium of the University of Santiago de Compostela and housed under conditions of controlled temperature (23 °C) and illumination (12/12 h light/dark cycle), with ad libitum access to standard pelleted food and tap water. All experiments and procedures were reviewed and approved by the Ethics Committee of the Universities of Cordoba (experiments involving rats) and Santiago de Compostela (experiments involving mice), and were conducted in accordance with European Union guidelines for the use of experimental animals. In addition, we used Kiss1[Cre:GFP] mice[49] at OHSU, for single-cell PCR analyses. Animal procedures related with these studies were approved by the Oregon National Primate Research Center (ONPRC) Animal Care and Use Committee, and were conducted in accordance with established NIH guidelines.

**Models of nutritional stress**. To address the putative role of hypothalamic SIRT1 in the metabolic control of puberty, we used models of perturbed puberty due to metabolic stress, caused by conditions of persistent energy excess or deficit. To generate a model of early overnutrition (ON), causing overweight, female pups were cross-fostered and raised in small litters (SL) of 4 pups/dam from PND1 onwards. Normal litters (NL) of 12 pups per dam served as a model of normal nutrition (NN). After weaning, SL rats were fed with a high-fat diet from PND21 to PND36 (HFD; Diet D12451, 45% fat content; Research Diets Inc., New Brunswick,

New Jersey) to induce early-onset obesity[50]. NL rats were fed after weaning with a control diet (Diet D12450B, 10% fat content). A model of peripubertal undernutrition (UN) was also generated; female rats reared as NL and fed the control diet were subjected to a 25% reduction in daily food intake from weaning (PND21) to PND36, as described earlier[50]. NL rats fed with the above-described control diet served as controls given an NN.

**Phenotypic evaluation of pubertal maturation**. Somatic and reproductive indices of pubertal development were evaluated as previously described[50], including (a) body weight (BW) gain; (b) age of vaginal opening (VO), a consensus external marker of puberty in female rodents; (c) uterine and ovarian weight; and (d) serum LH and FSH levels. Based on previous data on the normal timing of puberty in female rodents[50–52], VO was monitored daily from PND25 until termination of the experiment, time at which uterine and ovarian weight were recorded and serum hormone levels were assayed. Further assessment of the precise age of completion of puberty was achieved by histological analysis of the ovary, as described below.

**Ovarian histological analysis for precise assessment of the completion of puberty**. Ovaries (including the oviduct and the tip of the uterine horn) were fixed for at least 24 h in Bouin solution, before dehydration and embedding in paraffin. Serial (7 μm) sections were stained with hematoxylin and eosin, and evaluated under a microscope. Pubertal progression was estimated using a scoring method (Pub-Score), recently validated by one of our groups, based on histometric analyses of follicular development and development of corporal lutea[52]. This method dates pubertal maturation based on the combined analysis of follicular development and corpus luteum dynamics (the latter, for animals that has completed first ovulation). In nonovulating animals, the most advanced healthy antral follicle class, from small follicles measuring less than 275 μm in diameter to antral follicles (from F1 to F4) was determined, allowing to date prepubertal maturation from stage −4 to −1 (representing the interval expected between the age at analysis and the first ovulation). In addition, for animals that had undergone ovulation, dating of CL, as a morphological sign of ovulation, was also implemented, based on major histological features, allowing staging of pubertal timing at one-day intervals, from +1 (equivalent to CL1) to +4 (equivalent to CL4). Negative scores denote expected days until first ovulation, whereas positive scores indicate days after the first ovulation, therefore providing an integral assessment of the stage of pubertal maturation, even in animals that have not completed puberty.

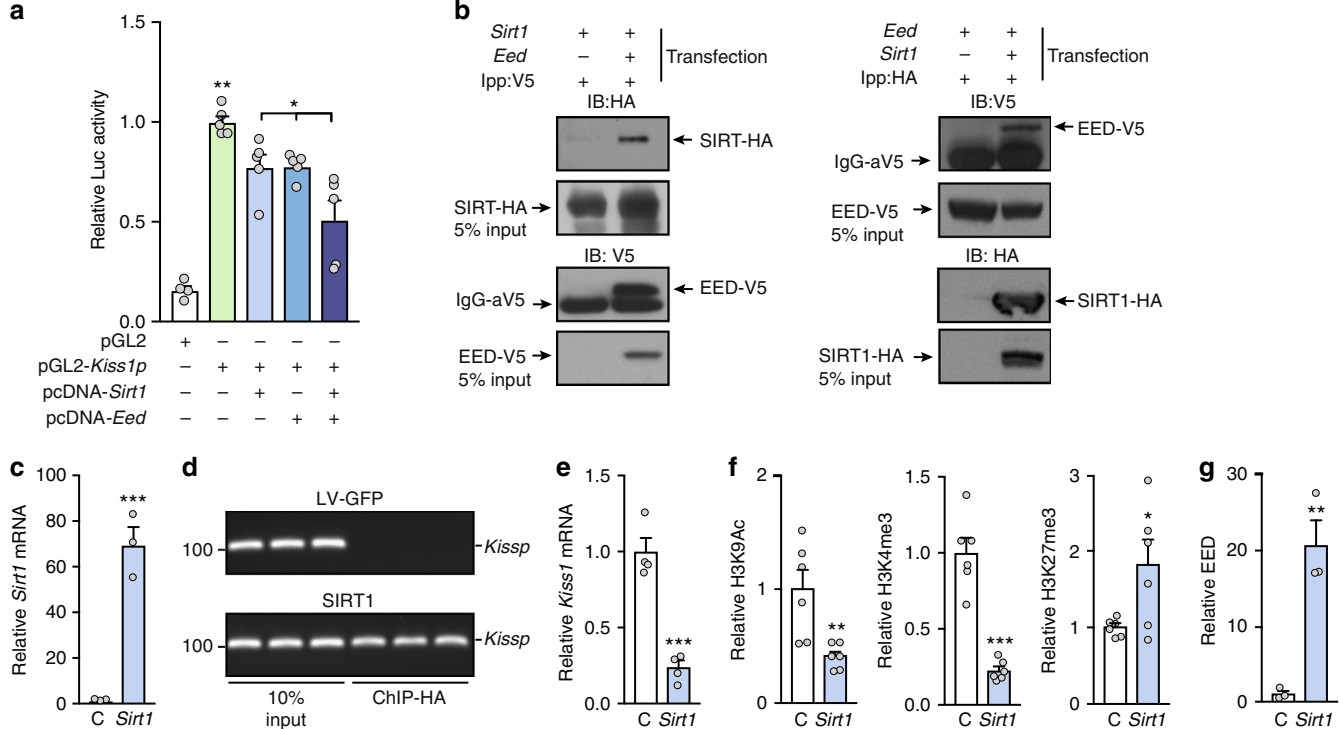

**Fig. 8** Kiss1 expression is repressed by an SIRT1-EED silencing complex. **a** SIRT1 and EED repress *KISS1* promoter activity as measured by gene reporter assays performed in Neuro2A cells. The bar histograms represent the mean ± SEM. *$p < 0.05$; **$p < 0.01$ (one-way ANOVA followed by the Student −Newman−Keuls; $n = 4$ biological replicates per group). **b** SIRT1 and EED coimmunoprecipitate as determined in 293T cells transfected with SIRT1-HA and EED-V5 expression vectors; reciprocal pull-down assays are shown. **c** Sirt1 mRNA in rat hypothalamic R22 cells infected with a control (C) lentiviral construct expressing green fluorescent protein (LV-GFP) or SIRT1. **d** CHIP assay showing association of SIRT1-HA to the *Kiss1* promoter. **e** Kiss1 mRNA in C and SIRT1 overexpressing hypothalamic R22 cells. **f** Detection of H3K9Ac, H3K4me3 and H3K27me3 in C or SIRT1 overexpressing R22 cells. **g** Recruitment of EED to the *Kiss1* promoter in C and SIRT1 overexpressing R22 cells. The bar histograms represent the mean ± SEM. *$p < 0.05$; **$p < 0.01$; ***$p < 0.001$ vs. C group (two-sided Student's *t* test; $n = 4$ per group)

**Surgical procedures**. For intracerebral injections, standard procedures for cannulation of the lateral cerebral ventricle, followed by chronic intracerebroventricular (icv) administration, were carried out using previously published protocols[53,54]. In brief, to access the lateral cerebral ventricle, the cannulae were lowered to a depth of 3 mm beneath the surface of the skull, with an insert point that was 1 mm posterior and 1.2 mm lateral to bregma. Immature female rats received icv injections of the allosteric SIRT1 activator, Sirt1-Activator 3 (SA3; Cayman Chemical Co.)[28,33], at a dose of 5 nmol, twice a day between PND26 and PND35. Pair-aged rats injected with vehicle (VH; 40% dimethyl sulfoxide in 0.9% sodium chloride) served as controls.

**Tissue collection**. For expression analyses, hypothalamic tissue containing the POA and the MBH was dissected immediately upon decapitation of the animals, by a horizontal cut ∼ 2 mm in depth with the following boundaries: one cut made 1 mm anterior of the optic chiasm, a posterior cut made at the border of the mammillary bodies, and two lateral cuts made at the hypothalamic sulci[53,55]. In addition, expression analyses were conducted on the POA, containing the AVPV region, and the MBH, which includes the ARC; dissection of these regions was carried out as previously recommended[56], with modifications. In brief, from the above rostral limit (1 mm anterior to the rostral chiasm), the POA region was dissected by a second cut immediately caudal to the optic chiasm, and two additional lateral cuts along the borders of the optic chiasm and the border of the anterior commissure. The distal hypothalamic fragment corresponded to the MBH, delimited by lateral cuts along the hypothalamic sulci and the mammillary bodies caudally. The tissue fragments were frozen in liquid nitrogen and stored at −80 °C until used for further analysis.

**Hormone assays**. Serum LH and FSH levels were measured using RIA kits supplied by the National Institutes of Health (Dr. A.F. Parlow, National Hormone and Peptide Program, Torrance, CA). Rat LH-I-10 and FSH-I-9 were labeled with $^{125}I$ by the chloramine-T method, and hormone concentrations were expressed using reference preparations LH-RP-3 and FSH-RP-2 as standards. Intra- and inter-assay coefficients of variation were less than 8 and 10% for LH and 6 and 9% for FSH,

respectively. The sensitivity of the assay was 5 pg/tube for LH and 20 pg/tube for FSH. Accuracy of hormone determinations was confirmed by assessment of rat serum samples of known concentrations, used as external controls.

**Western blots**. Total protein was extracted from hypothalamic blocks or specific hypothalamic areas (POA and MBH), following previously described procedures[57]. Briefly, total protein lysates (20 μg) were subjected to SDS-PAGE on 7−12 % polyacrylamide gels, electro-transferred on polyvinylidene difluoride (PVDF) membranes (Millipore) and probed overnight at 4 °C in the presence of the following primary antibodies: anti-SIRT1 (1:1000; 8469S, Cell Signaling), anti-histone 3 (anti-H3; 1:5000 dilution, 06-755, Millipore) and anti-acetyl histone 3 lysine 9/14 (anti-H3K9/14ac; 1:10,000 dilution, 06-599, Millipore). For protein detection, we used horseradish peroxidase-conjugated secondary antibodies and chemiluminescence (Abcam). Six samples per group were assayed; protein levels were normalized to β-actin (1:5000 dilution, A5060, Sigma Aldrich). A summary of the primary antibodies and dilutions used in western blot analyses is provided in Supplementary Table 1. Densitometry analysis of protein bands was conducted using the open source image processing software, ImageJ (https://imagej.net/ImageJ). Uncropped scans of relevant western blots included in this work are presented as Supplementary Figure 5.

**RNA extraction, reverse transcription and quantitative (q)PCR**. Total RNA was extracted from hypothalamic tissues and cultured cells using the RNeasy mini kit (Qiagen, Valencia, CA) or TRI-sure isolation reagent (Bioline Reagents Ltd., UK) following the manufacturer's instructions. For Sirt1 and Kiss1 mRNA measurements in cultured cells, we treated LV-*Sirt1-HA*-infected R22 (see below) cells with 5-azacytidine (Aza; 1 nM; Sigma-Aldrich) to reduce DNA methylation-dependent repression of gene expression. The treatment was initiated 48 h before mRNA extraction. RNA concentrations were determined by spectrophotometric trace (Nanodrop, ThermoScientific, Wilmington, DE). Total RNA (1000 ng) was transcribed into cDNA in a volume of 20 μl using 4U Omniscript reverse transcriptase (Qiagen) or the IScript cDNA Synthesis kit (Bio-Rad Laboratories Inc., USA). To determine the relative abundance of the mRNAs of interest, we used the SYBR

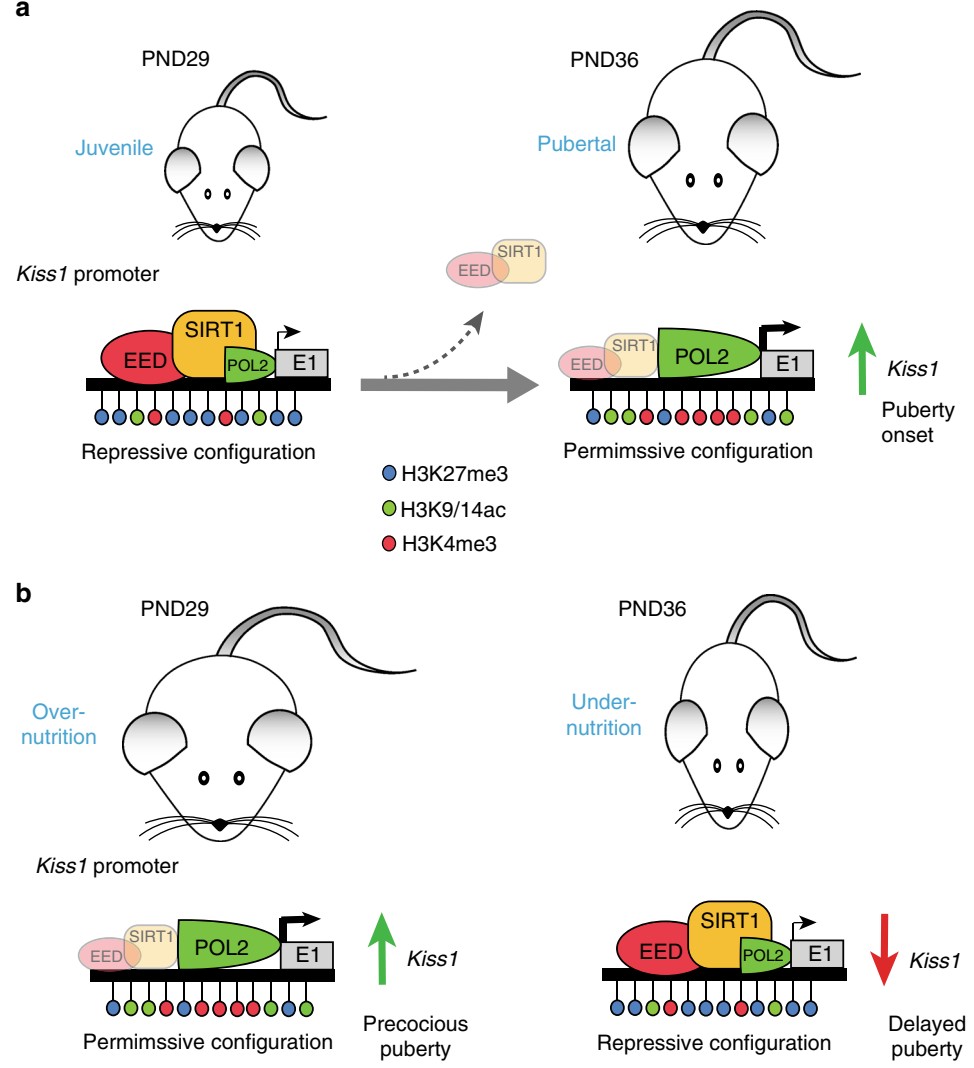

**Fig. 9** Mode of SIRT1 action in the control of puberty and its modulation by metabolic cues. **a** Major events occurring during normal female pubertal maturation. Transition from late juvenile (PND29; left panel) to peripubertal (PND36; right panel) stages is defined by eviction of SIRT1 and EED, a key member of the PcG silencing complex, from the *Kiss1* promoter in KNDy neurons, which changes the chromatin landscape from a predominantly repressive to a permissive histone configuration. According to this model, these changes would result in enhanced *Kiss1* transcription, mandatory for puberty onset. **b** Predicted changes in *Kiss1* transcriptional activity occurring under opposite nutritional conditions. In the left panel, early-onset obesity (caused by overnutrition) induces the premature eviction of SIRT1/EED from the *Kiss1* promoter, and the rearrangement of histone configuration from repressive to permissive. This change, already evident by PND29, would allow enhancement of *Kiss1* transcription, leading to precocious puberty. In contrast, prepubertal under-nutrition (right panel) prevents the eviction of SIRT1/EED from the *Kiss1* promoter, which maintains a repressive histone configuration still at PND36. The model predicts that this protracted repression would lead to decreased *Kiss1* transcription and delayed puberty

GreenER™ qPCR SuperMix system (Invitrogen, Carlsbad, CA) or Go-Taq qPCR Master mix (Promega Corporation, USA). Primers used were as follows: (a) r*Kiss1*; accession number NM_181692.1, forward1 5′-GCTGCTGCTTCTCCTCTGTG-3′ and reverse1 5′-GCATAC CGCGGGCCCTTTT-3′, used in experiments shown in Figs. 1, 2 and 4; and forward2 5′-TGGTGAACCCTG AACCCACAGGC-3′ and reverse2 5′-CGGCGGGCATGGCGATGTT-3′ used in experiments shown in Fig. 8; (b) m*Sirt1*, accession number NM_019812, forward1 5′-GACAGAACGTCA CACGCCAG-3′ and reverse1 5′-TTGTTCGAGGATCGGTGCCAA-3′, used in experiments shown in Fig. 3; (c) r*Tac3*; accession number NM_019162.2, forward1 5′-CAGCTTGGCATGGACCTTC-3′ and reverse1 5′-TAGCCTTGCTCAGCA C TTTCA-3′. PCR reactions were performed in a total volume of 10 μl containing 1 μl of diluted cDNA or a reference cDNA sample (see below), 5 μl of SYBR GreenER™ qPCR SuperMix and 4 μl of primers mix (1 μM of each gene specific primer). The PCR conditions used were 95 °C for 5 min, followed by 40 cycles of 15 s at 95 °C and 60 s at 60 °C. Cycles (CTs) were detected by QuantStudio 12K Flex software. To confirm the formation of a single SYBR Green-labeled PCR amplicon, the PCR reaction was followed by a three-step melting curve analysis consisting of 15 s at 95 °C, 1 min at 60 °C, ramping up to 95 °C at 0.5 °C/s, detecting every 0.5 s and finishing for 15 s at 95 °C, as recommended by the manufacturer.

Relative standard curves were constructed from serial dilutions (1/2 to 1/1,024) of a pool of cDNAs generated by mixing equal amounts of cDNA from each sample. The CTs from each sample were referred to the relative standard curve to estimate the mRNA content/sample. The values obtained were normalized for procedural losses using the mRNA encoding the ribosomal protein S11 (*Rps11*), accession number NM_031110.1, forward 5′-CATTCAGACGGAGCGTGCTTA-3′ and reverse 5′-TGCATCTTCATCTTCGTCAC-3′, glycer-aldehyde-3-phosphate dehy-drogenase (*GAPDH*), accession number XM_017593963, forward 5′-CTTCTT GTGCAGTGCCAGCC-3′ and reverse 5′-CAAGAGAAGGCAGCCCTGGT-3′, or peptidyl-prolyl isomerase A (*Ppia*), accession number M19533.1, forward 5′-GGCAAATGCTGGACCAAACACAA-3′ and reverse 5′-GGT AAAATGCCC GCAAGTCAAAGA-3′, as the normalizing units. A summary of primers used in qPCR assays is provided in Supplementary Table 2.

**Virogenetic overexpression of SIRT1 in the ARC**. Using a stereotaxic approach[58,59], immature female rats (PND20) were injected bilaterally into the ARC with an adeno-associated virus (AAV) vector expressing green fluorescent protein (GFP; AAV-C) or SIRT1 (AAV-SIRT1). To this end, rats were placed in a

stereotaxic frame (David Kopf Instruments) under ketamine/ xylazine anesthesia. The injections were targeted to the ARC (stereotaxic coordinates from Bregma AP: −2 mm; L: ±0.3 mm; DV: −9.4 mm) using a 33-gauge Nano-Fil needle (WPI) connected to a Nano-Fil 10-μl syringe (WPI). AAV vectors expressing GFP (1× 1010 pfu/ml; Tebu-Bio) or SIRT1 (1×10$^{11}$ pfu/ml; Tebu-Bio) were delivered at a rate of 200 nl/min for 1 min (1 μl/ injection site) and the entire injector system was left in place for an additional 5 min after the injections were completed. Phenotypic monitoring of puberty included assessment of VO, which was monitored daily from PND28 to 36, and occurrence of the first estrus (after VO). On PND36, animals were euthanized, body, ovarian and uterus weights were recorded, and brain samples were collected to visualize the location of the injections by fluorescein isothiocyanate (Sigma Aldrich) or fluorescence latex microspheres (Lumafluor Inc), coadministered with AAV injections, both in AAV-C and AAV-SIRT1 animals, while assessment of GFP was used as marker of effective infection. Immunohistofluorescence localization of GFP was conducted according to previously validated protocols, using a polyclonal antibody against GFP (Abcam; 1/ 2000 dilution)[58]. In addition, in a subset of AAV-C and AAV-SIRT1 animals, MBH tissue samples were collected upon euthanasia to measure SIRT1 and histone levels by western blots (as described in the section Western blots).

**EED-SIRT1 coimmunoprecipitation assay.** To determine EED–SIRT1 protein −protein interaction, we first cloned the coding region of rat *Eed* using the primers (accession number NM_001106278.2, forward 5′-AATAGGATCCAC-CATGTCTGAGAGGGAAGT GTCGACTGCG-3′ and reverse 5′-TACTGAT ATCTCGAAGTCGATCCCAACGCCAAATGCT-3′) into pcDNA3.1/V5-His(B) (Invitrogen) at the *Bam*HI/*Eco*RV sites, generating the expression vector pcDNA3.1-rEED-V5-His. Mouse *Sirt1*-FLAG-HA was obtained by digesting the pcDNA3-FLAG-HA-*Sirt1* (provided by Dr. Michael Cowley, Monash University, Melbourne) with *Nru*I, blunted with T4 DNA polymerase followed by digestion with *Xba*I. The lentiviral vector pLVIIRES was digested using *Cla*I, blunted with T4 DNA polymerase followed by digestion with *Xba*I. Both fragments were ligated with T4 DNA ligase producing the vector pLVIIRES-Sirt1-FLAG-HA. For coimmunoprecipitation assays, we seeded 1.2×10$^7$ HEK293T cells (HEK 293T/17, CRL-11268$^{TM}$; from ATCC, Manassas, VA) in 10 cm plates in DMEM supplemented with 10% fetal bovine serum (FBS). The following day the cells were transfected with either pcDNA3.1-rEED-V5-His, LVI-Flag-HA-m*Sirt1* or both at 10 μg DNA and 25 μl Lipofectamine 2000 per plate in Optimem (Invitrogen). After 5 h, the mix was replaced with complete DMEM medium-10% FBS. Twenty-four hours later cells were washed twice with PBS and resuspended in lysis buffer consisting of 25 mM Tris (pH 7.4), 150 mM NaCl, 1 mM EDTA, 1 mM EGTA, 1% Triton X-100 containing a protease inhibitor cocktail (1 mM phenylmethylsulfonyl-fluoride, 7 μg/ml aprotinin, 0.7 μg/ml pepstatin A, 0.5 μg/ml leupeptin) and a phosphatase inhibitor cocktail (2.5 mM Sodium pyrophosphate, 1 mM β-Glycerol-phosphate, 1 mM Na$_3$VO$_4$ and 1 mM sodium fluoride). The lysates were homogenized by passage through a 27 g syringe 4−6 times and insoluble proteins were pelleted by centrifugation for 10 min at 1000 × *g*. The lysates were precleared with agarose conjugated protein A (Sigma) and subsequently incubated with 3 μg of V5 antibody (R960-25, Invitrogen) or 3 μg of HA antibody (ab9110, Abcam) overnight at 4 °C with 50 μl agarose conjugated protein A. Following immunoprecipitation, all samples were washed five times with lysis buffer before the agarose beads were resuspended in SDS-sample buffer. After boiling, the supernatants were subjected to SDS-PAGE, transferred to PVDF membranes (Millipore), blocked in 5% nonfat dry milk and probed for either V5 or HA epitopes with the antibodies described above diluted 1/2000 and detected with horseradish peroxidase-labeled secondary antibodies (81-6120, Zymed Laboratories) diluted at 1/25,000. All blots were developed following standard procedures, as reported previously[26,58].

**Functional promoter assays.** To determine if SIRT1, alone or with EED, alters the transcriptional activity of *KISS1*, we transfected Neuro2A cells (N2A, from ATCC) with a luciferase reporter construct (pGL2-Kiss1 Luc) containing the transcription start site (TSS) and adjacent sequence of human *KISS1* (accession number NM_002256.3, 1339 bp sequence, from −1317 to +22). Cells were maintained in Dulbecco's modified Eagles medium containing high glucose (4.5 g/l; Sigma), supplemented with 10% FBS (Invitrogen), ʟ-glutamine (2.5 mM, Sigma), 100 U/ml penicillin, and 100 μg/ml streptomycin (Invitrogen) and cultured at 37 °C in a humidified atmosphere containing 5% CO$_2$. For promoter assays, the cells (400,000 cells/well) were seeded onto 24-well plates in DMEM containing 10% FBS. Twenty-four hours later, the *KISS1* reporter construct (100 ng per well) was transiently transfected alone or cotransfected with pcDNA3-FLAG-HA-*Sirt1*, pcDNA3.1-rEED-V5-His or both (100 ng per well), using Lipofectamine 2000 (Invitrogen) at a ratio of 1 μg DNA: 2.5 μl Lipofectamine 2000 in Optimem (Invitrogen). After 5 h of transfection, the cells were returned to serum-containing DMEM medium. Forty-eight hours later, the cells were harvested and assayed for luciferase activity using the Firefly Luciferase Glow Assay Kit (Pierce, Rockford, IL). The assay was performed in opaque 96-well plates and light emission measured in a Spectramax M5 microplate reader (Molecular Devices, Sunnyvale, CA). Transfection efficiency was normalized by cotransfecting the plasmid CMV-Sport-β–gal (Invitrogen) at 10 ng per well, and determining β-Galactosidase activity using the Tropix Galacto Light Plus (ABI), as reported earlier[59].

**Chromatin immunoprecipitation (ChIP) assays.** To assess the recruitment of SIRT1 to specific gene promoters, and the association of different histone modifications to these promoters in vivo, we performed ChIP assays using chromatin extracted from the MBH and POA of prepubertal NN, UN (PND29 and PND36), and ON (PND29) female rats. Because initial analyses revealed only modest changes at *Kiss1* promoter in POA samples, ChIP assays at this site were restricted to the assessment of interactions with SIRT1, H3K9Ac, and H4K16Ac. To assess the changes in histone modifications resulting from *Sirt1* overexpression, ChIP assays were performed using chromatin extracted from embryonic rat hypothalamic cells, R22 (Cedarlane Laboratories, Canada), infected with either LV-*Sirt1*-HA or LV-GFP constructs. ChIP procedures were described previously by one of our groups[26,60,61], and were carried out with minimal modifications, as follows. Cells were harvested for chromatin immunoprecipitation 48 to 72 h after infection. The cells and tissue fragments were washed once in ice-cold PBS containing a protease inhibitor cocktail (PI, 1 mM phenylmethyl-sulfonylfluoride, 7 μg/ml aprotinin, 0.7 μg/ml pepstatin A, 0.5 μg/ml leupeptin), a phosphatase inhibitor cocktail (PhI, 1 mM β-glycerophosphate, 1 mM sodium pyrophosphate and 1 mM sodium fluoride), and an HDAC inhibitor (20 mM sodium butyrate). Thereafter, cells and tissue fragments were cross-linked by exposing them to 1% formaldehyde for 10 min at room temperature. After two additional washing steps in PBS the samples were lysed with 200 μl SDS buffer (0.5% SDS, 50 mM Tris-Hcl, 10 mM EDTA) containing protease, phosphatase, and HDAC inhibitors and sonicated for 45 s to yield chromatin fragments of approximately 500 base pairs (bp) using the microtip of a Fisher Scientific FB 705 sonicator. Size fragmentation was confirmed by agarose gel electrophoresis. The sonicated chromatin was clarified by centrifugation at 14,000 rpm for 10 min at 4 °C, brought up to 1 ml in Chip Dilution Buffer (16.7 mM Tris-HCl, pH 8.1, 150 mM NaCl, 1.2 mM EDTA, 1.1% Triton X-100, and 0.01% SDS) containing the PI and PhI cocktails, and the HDAC inhibitor described above. The samples were then stored at −80 °C for subsequent immunoprecipitation. For this step, chromatin was precleared with Protein A/G beads (Dynabeads, Invitrogen, Carlsbad, CA) for 1 h at 4 °C. For chromatin immunoprecipitation, 25−50 μl aliquots of chromatin were incubated with 2−5 μg of various antibodies: SIRT1 (sc-15404, Santa Cruz); H3K4me3 (39159, Active Motif); H3K9ac, (39917, Active Motif); H3K27me3 (39155, Active Motif); H4K16ac (39929, Active Motif); or Beta-Galactosidase (CR7001RP2, Cortex Biochem). A list of the antibodies used in ChIP assays is provided in Supplementary Table 1. Antibody-chromatin complexes and 25 μl of protein A or G beads solution (Dynabeads) were incubated at 4 °C overnight with gentle agitation. Immuno-complexes were washed sequentially with 0.5 ml low salt wash buffer (20 mM Tris-HCl, pH 8.1, 150 mM NaCl, 2 mM EDTA, 1% Triton X-100 and 0.1% SDS), high salt wash buffer (20 mM Tris-HCl, pH 8.1, 500 mM NaCl, 2 mM EDTA, 1% Triton X-100 and 0.1% SDS), LiCl buffer (10 mM Tris-HCl, pH 8.1, 250 M LiCl, 1% Nonidet P-40, 1% sodium deoxycholate and 1 mM EDTA), and TE buffer (10 mM Tris-HCl, pH 8.0 and 1 mM EDTA). The immunocomplexes were eluted with 100 μl of 0.1 M NaHCO$_3$ and 1% SDS at 65 °C for 45 min. Cross-linking was reversed by adding 4 μl of 5 M NaCl and incubating at 95 °C for 30 min. DNA was recovered by using ChIP DNA Clean & Concentrator columns (Zymo Research, Irvine, CA) and stored at −80 °C before PCR analysis. Unless otherwise stated, all chemicals for ChIP assays were purchased from Sigma-Aldrich (St. Louis, MO, USA).

**qPCR detection of chromatin immunoprecipitated DNA.** Genomic regions of interest were amplified by qPCR. Accession numbers of the genes analyzed as well as the chromosomal position of the 5′-flanking region amplified, using the position of the TSS as the reference point, were as follows: r*Kiss1* (accession number NM_181692.1, 91 bp product), r*TAC3* (accession number NM_019162.2, 123 bp product) and r*ProDyn* (accession number NM_019374.3, 109 bp product). Primer sequences used were: r*Kiss1* (forward3 5′-TCGGGCAGCCAGATAGAGGAAGC -3′; reverse3 5′-TTGAGGGCCGAGGGGAGAAGAG-3′), r*Tac3* (forward2 5′-ACGT GCGTGTCTGGGTATGTGA-3′; reverse2 5′-GGAGGGTTTGGGGGAGTCG-3′) and r*ProDyn* (forward 5′-CTGCCTTTCTCCTACTTTTGT CTCTGTTTT-3′; reverse: 5′-CGGGGGTGGATTCTCGGTGTAG-3′). A summary of primers used in ChIP assays is provided in Supplementary Table 2. PCR reactions were performed using 1 μl of each immunoprecipitate (IP) or input samples (see below), primer mix (1 μM each primer), and SYBR Green Power Up Master Mix™ (Thermo Fisher, Waltham, MA) in a final volume of 10 μl. Input samples consisted of 10% of the chromatin volume used for immunoprecipitation. The thermo-cycling conditions used were: 95 °C for 5 min, followed by 40 cycles of 15 s at 95 °C and 60 s at 60 °C. Data are expressed as % of IP signal/Input signal.

**Detection of *Sirt1* expression in GFP-expressing kisspeptin neurons by single-cell-RT-PCR.** To better visualize kisspeptin neurons of the ARC, we used 1-week ovariectomized (OVX) female Kiss1$^{Cre:GFP}$ mice (*n* = 4 animals). Single cells were dispersed and harvested according to established procedures[49], with minor modifications, as follows. Briefly, the ARC was micro-dissected from coronal slices obtained from Kiss1$^{Cre:GFP}$ mice. Following extensive washing in oxygenated artificial CSF, the dispersed cells were visualized using a Leitz inverted fluorescent microscope, patched, and then harvested with gentle suction to the pipette using a Xenoworks manipulator system (Sutter Instrument; Novato, CA). The cells were expelled into a siliconized 0.65-ml microcentrifuge tube containing a solution of 1× Invitrogen Superscript III Buffer, 15 U of RNasin (Promega), 10 mM dithiothreitol

and diethylpyrocarbonate-treated water in a total of 5 µl for a single cell. cDNA synthesis was performed on single cells as previously described[62], and stored at −20 °C. Controls were prepared to detect contamination in both the harvesting procedure and the PCR. Artificial CSF (aCSF) was collected from the plate in the vicinity of the dispersed cells as a negative control. Single cells were harvested and taken through the reverse transcription but without RT enzyme to confirm that there was no amplification of genomic DNA. In the PCR procedure, we included tissue controls consisting of 25 ng of basal hypothalamic RNA that were reverse transcribed with and without RT enzyme as a positive and negative control. A water blank was also included in the PCR that contained the master mix for the PCR but no cDNA as a negative control.

Primers for Kiss1 and Sirt1 amplification were designed using Clone Manager software (Sci Ed Software), to cross at least one intron−exon boundary and optimized as previously described[62]. The primers used were as follows: Sirt1 (accession number NM_019812; 198-bp product; forward primer 694−711 nt; reverse primer 872−891 nt), Kiss1 (accession number NM_178260; 120-bp product; forward primer 64−80 nt; reverse primer 167−183 nt). Details of the primers used for single-cell RT-PCR assays (including sequences) are provided in Supplementary Table 2. The PCR was performed using 2−3 µl of cDNA template from each RT reaction in a 30 µl PCR mix as follows: initial denaturation for 2 min at 94 °C; 20 s at 94 °C, 30 s at 61 °C (Sirt1) or 57 °C (Kiss1), 30 s at 72 °C for 50 cycles of amplification, with a final extension of 5 min at 72 °C. PCR products were sequenced to confirm their identity.

**Quantitation of SIRT1 in kisspeptin neurons by immunhistofluorescence**.
Animals were intracardially perfused with 4% paraformaldehyde in PBS. After removing the brains, they were submerged in the same solution overnight at 4 °C followed by two days in 30% sucrose in PBS. Thirty micron slices were made on a sliding cryostat, mounted into microscope slides and vacuum-dried overnight at room temperature. The sections were incubated overnight at 4 °C with a rabbit polyclonal antibody against kisspeptin (Ab9754 from Millipore, diluted 1:2000), followed by biotyramine enhancement method, before developing the reaction to a red color with Strepavidin Alexa 568 (Invitrogen; dilution 1:500). Next, sections were incubated with anti-SIRT1 antibodies (sc-74504, Santa Cruz Biotechnologies, Dallas, TX; 1:1000 dilution), and the reaction was developed the next day to a green color using Alexa 488 donkey anti-goat IgG (Invitrogen, 1:500). Fluorescent images were acquired with a Zeiss Axiovert 200 M microscope with a ×20 C-apochromat NA1.2 objective. Kisspeptin cells were identified as immunoreactive neuron-like cells that stood out above background, and that were clearly discernible from immunoreactive fibers. To quantify changes in cellular SIRT1 immunoreactivity, each kisspeptin cell was manually outlined and the intensity of the SIRT1 signal was measured using ImageJ software. Each outline was then moved over neighboring nonlabeled areas to obtain a measure of the mean intensity of the background. Following subtraction of the background values from each cell, the mean intensity values per group were calculated.

**Fluorescent in situ hybridization (FISH)**. Brains from PND28 female rats ($n = 3$) were fixed by intracardiac perfusion of 4% paraformaldehyde borate buffer, and were processed for hybridization histochemistry, as previously described[63]. In detail, we used the double FISH procedure employing nonradioactive complementary (c)RNA probes, as optimized by us[26]. A cRNA complementary to Sirt1 mRNA was labeled with fluorescein-12-UTP. A Kiss1 cRNA probe was labeled with digoxigenin-11-UTP. The labeling reactions were performed in a 10 µl volume, as reported[64]. Control sections were incubated with sense probes transcribed from the same plasmid, but linearized on the 3′ end to transcribe the coding strand of the cDNA template.

**Probes for in situ hybridization**. The Kiss1 cRNA probe was transcribed from a 393 bp rat Kiss1 cDNA corresponding to nucleotides (nt) 1-393 in rat Kiss1 mRNA (accession number NM_181692.1, forward4 5′-ATGATCTCGCTGGCTTCTTGGCAG-3′ and reverse4 5′-TCAGCCCCGTGCCGCCCGCGC-3′). A rat Sirt1 cRNA probe was generated by in vitro transcription of a 569 bp cDNA template complementary to nt 29−598 in the coding region of rat Sirt1 mRNA (accession number XM_017588054.1, forward3 5′-ATGACAGAGCATCACACGCA-3′ and reverse3 5′-CTGCAACCTGCTCCAAGGTA-3′). All RNA probes were prepared by in vitro transcription of cDNA templates generated by RT-PCR amplification of total RNA extracted from the MBH of 28-day-old female rats. PCR fragments were cloned into the pGEM-T vector (Promega) and verified by sequencing.

**Statistics**. Statistical analyses were performed using Prism software (Graphpad Prism version 7.00 for Windows, GraphPad Software, La Jolla, California, USA, www.graphpad.com). The data were first subjected to normality tests. For data passing these tests, the differences between several groups were analyzed by one-way ANOVA followed by the Student−Newman−Keuls multiple comparison test for unequal replications. The Student's $t$ test was used to compare two groups. Analyses were two-sided. Quantitative, continuous data are expressed as the mean ± SEM for each group. When comparing percentages, groups were subjected to arc−sine transformation before statistical analysis to convert them from a binomial to a normal distribution; statistical differences of frequency distributions were

calculated using $X^2$ tests. A P value of <0.05 was considered statistically significant. The sample size was selected based on previous experience with studies addressing molecular and neuroendocrine regulation of puberty, assisted by power analyses performed using values of standard deviation that we usually obtain when measuring parameters analogous to those examined in this study. Based on those calculations, a minimal group size of $n = 6$ animal per group was established as a general rule, as analyses using these sample size should provide at least 80% power to detect effect sizes using the tests indicated above, with a significance level of 0.05. Note that for physiological experiments, groups sizes largely exceeded this threshold; yet, based on standard procedures, while phenotypic and hormonal analyses were applied to all individuals, more complex molecular and histological analyses in these experiments were implemented in a representative subset of randomly assigned samples from each group. Further details are provided in the corresponding figure legends. For molecular (ChiP) assays, tissues from four individuals per experimental group were studied; assays were done in duplicate. As a general rule, the investigators directly performing the experimentation involving physiological/molecular determinations were not blinded to the group allocation, but primary data analyses conducted by senior authors were conducted independently to avoid any potential bias.

### Data availability
The authors declare that the data supporting the findings of this study are included in this published Article and its Supplementary Information files. All relevant original data are available from the corresponding authors, upon reasonable request.

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

## Acknowledgements

This article is dedicated to the memory of late Prof. Enrique Aguilar. This work was supported by grants BFU2011-025021, BFU2014-57581-P, and BFU2017-83934-P (Ministerio de Economía y Competitividad, Spain; co-funded with EU funds from FEDER Program) to M.T.-S.; project PIE-00005 (Flexi-Met, Instituto de Salud Carlos III, Ministerio de Sanidad, Spain) to M.T.-S.; Projects P08-CVI-03788 and P12-FQM-01943 (Junta de Andalucía, Spain) to M.T.-S.; EU research contract DEER FP7-ENV-2007-1 to M.T.-S.; and the National Institute of Health (1R01HD084542) to S.R.O. and A.L., (RO1DK068098) to O.K.R. and 8P51OD011092 for the operation of the Oregon National Primate Research Center. This project has received funding from the EU Horizon 2020 Program, under the Marie Sklodowska-Curie grant agreement No. GAP-2014-655232. C. A.T. was supported by NIH Training grant T32-HD007133 and NRSA grant F32-HD-86904, and J.M.C. was funded by SAF2014-56995-JIN (Ministerio de Economía y Competitividad, Spain). CIBER Fisiopatología de la Obesidad y Nutrición is an initiative of Instituto de Salud Carlos III. Senior authors are indebted to M. Serrano (IRB, Barcelona, Spain) for provision of relevant mouse lines, essential for conduction of some of the experiments of this study.

## Author contributions

M.J.V. participated in study design, conducted and coordinated the physiological experiments, evaluated and discussed the data, assisted in figure and manuscript preparation; C.A.T. conducted molecular analyses, evaluated and discussed the data, and helped in preparing the figures—M.J.V. and C.A.T. should be considered joint first authors; J.M.C. participated in study design, assisted in conduction of experimental studies, discussed the data and assisted in manuscript preparation; F.R.-P. actively participated in the conduction of the various experimental studies and analytical procedures, with the support and assistance of J.R., V.H., and I.V., who all discussed the data; D.B., C.D. and R.N. conducted experimental studies in GM mouse lines and discussed the data; M.A.B. and O.K.R. did the single-cell RT-PCR experiments, L.P. and F.G. participated in study design and analytical procedures, and actively discussed the data; M.T.-S., S.R.O., and A.L. designed the study; A.L. led the molecular and localization analyses, and contributed to analyze the results; M.T.-S. led physiological studies, and also contributed to analyzing the data. S.R.O. and M.T.-S. wrote the manuscript, which was reviewed by A.L. and the rest of the authors. All the authors take

full responsibility for the work. J.M.C. and F.R.-P. equally contributed to this work. A.L. and M.T.-S. are both corresponding authors of this study.

## Additional information

**Competing interests:** The authors declare no competing interests.

