## [Peer Review File · Nature Communications]

Reviewers' comments:

Reviewer #1 (expert in nutritional sensing/sirtuins)(Remarks to the Author):

This is an extremely interesting study hypothesizing a specific epigenetic mechanism in certain hypothalamic neurons for the onset of puberty and its regulation by diet. This involves repression of the Kiss 1 gene by SIRT1 with the help of EED, leading to observed changes in histone marks at the promoter. SIRT1 transgenic mice and pharmacological SIRT1 activation are used to draw a causal relationship between SIRT1 at the Kiss1 promoter and delay of puberty. Since SIRT1 is a known nutrient sensor, the model can connect diet to onset of puberty in a satisfying way. Most of the experiments are well done and the data is convincing.

Several modifications would strengthen the paper. First, RNA analysis for Kiss1 and Tac3 should be provided for the studies in Fig. 1 through 4. This data is the basis of the hypothesis and should be complete. For example, one expects Kiss1 RNA to change predictably in all cases. Second, in Fig. 6a the H3K4me3 does not appear to correlate well with the hypothesis. I found this confusing and would require a more detailed explanation or deletion of the experiment altogether. Third, the reciprocal co-IP in Fig. 7b using HA to IP should be provided. Fourth, the authors should balance their discussion with the possibility that at least some of the effects in the hypothalamic neurons may not be cell autonomous. The transgenic mice or STAC injection are systemic and may exert effects on these neurons indirectly, eg. via adipokines. Granting that Fig. 7 in N2A cells is consistent with a cell autonomous model, only a brain specific SIRT1 transgenic mouse would rule out effects of WAT. A brain specific SIRT1 KO would also add to the study. However, I do not require additional experiments with new transgenic mice, which may take years. The authors should simply present a more sophisticated analysis of what their data actually show.

Less important, a few key references are missing. The fact that SIRT1 is a NAD-dependent deacetylase should be referenced as Imai et al., Nature 2000. The fact that SIRT1 transgenic mice show delayed puberty should be referenced by Bordone et al., Aging Cell 2007. This earlier study does not detract from the novelty of this paper, which is the epigenetic mechanism proposed.

In summary, this paper shows a molecular mechanism that may explain how diet influences the age of onset of puberty. It further suggests pharmacological ways to influence reproduction, which may have a place in population control. In this regard it would be fascinating to see whether NAD precursors restrain puberty. This study will be of broad interest to readers of Nat. Comm.

Reviewer #2 (expert in GnRH neurons and central regulation of reproduction)(Remarks to the Author):

Despite tremendous technical advances over recent decades, the far-reaching and apparently simple question of what initiates the onset of puberty continues to baffle endocrinologists. While this paper does not answer the question, it does identify what appears to be an important new link in the chain. Vazquez and Toro et al have shown that removal of the NAD-dependent deacetylase SIRT1 from the Kiss1 promoter modulates pubertal onset. As this conclusion seems robust, it follows logically that under- and over-nutrition effect SIRT1 levels on the Kiss1 promoter.... as these nutritional states are well documented to affect kiss1 levels and puberty onset. I am enthusiastic about this paper and think that it represents a significant advancement for the field. However, I have a number of questions which I think need to be addressed either here, or as discussion points.

1. In order to compliment the gain-of-function experiments, which importantly do not target Kiss1-neurons specifically, it would seem logical to generate mice with targeted deletion of SIRT1 in kiss1-neurons. The animals to do this are available from Jackson Labs. Did the authors attempt these experiments, or do they predict that an inducible Kiss1-cre would be required?
2. The manuscript focuses on the Arcuate hypothalamic nucleus. Is there evidence that SIRT1 action in the AVPV hypothalamus is also important in puberty onset?
3. Only approximately 50% of Kiss1 neurons express SIRT1 (Fig5). How does this fit in to the author's model of puberty onset? Do they predict higher Kiss1-expression, and a lack of involvement in puberty-onset, in pre-pubertal SIRT1-negative cells? I think this may need a

discussion point.

4. Did the authors look at SIRT1 levels before day 10 (Fig1a)? Is it possible that the animal is born with very high SIRT1 and that it is progressively lost, at a rate that is perhaps coupled to body-weight, until it reaches a level that is permissive for Kiss1 transcription?

5. Following on from point 4, the authors may want to speculate on future efforts to determine the nature of the signal that might initiate the removal of SIRT1 from the Kiss1 promotor.

In support of Nature Communications policy on manuscript review, I sign below.

Dr Bryn M Owen, PhD
Lecturer in Molecular Endocrinology
Imperial College, London.

Reviewer #3 (expert in obesity and kisspeptin)(Remarks to the Author):

This manuscript by Vazquez, Toro and co-workers examines the role of sirtuin 1 (sirt1) in metabolic regulation of puberty. They use established models of under and over nutrition to manipulate the timing of puberty and find enhancement and repression, respectively of sirt1 in the regions of the hypothalamus containing kisspeptin neurons, which are important upstream regulators of the pubertal process. Overexpression of Sirt1 did not alter body mass but did delay puberty in a manner similar to under nutrition; pharmacological activation of sirt1 had a similar effect. Kisspeptin neurons in the arcuate were shown to express Sirt1 and ChIP was used to demonstrate nutritional regulation of the chromatin of the Kiss1 and Tac2, but not Pdyn, genes. This is an interesting set of studies that nicely complement previous work demonstrating a role for epigenetic regulation of central processes in the initiation of puberty, and extending this work to physiologically-relevant models that alter the timing of puberty.

In figure 3, corresponding measures of Kiss1 mRNA and uterine mass (similar to figure 4) should be provided.

In figure 5, please provide quantification of the percent of Kiss1 neurons expressing Sirt1. Likewise, given the metabolic importance on non-kisspeptin cells in the arcuate nucleus, a quantification of Kiss1 negative/Sirt1 positive cells would be informative. Both of these measures would be stronger if an effect of metabolic condition was shown to regulate coexpression.

Also in figure 5, negative controls for single-cell PCR that include RT of the solution in which the harvested cells are suspended should be included.

The discussion nicely places the work in context, although it might be prudent to state that KNDy neurons are likely involved given the broad expression of Sirt1 in the arcuate and the lack of cell-specific modifications in the in vivo studies.

Line 246 Probably best to specify affiliation as here may be ambiguous.

Line 293 please give final concentration of DMSO injected.

Line 370 R22 cells are malignant melanoma cells, what is rationale for these when cell of interest is post mitotic?

Reviewer #4 (expert in brain epigenetics) (Remarks to the Author):

This is a potentially very interesting study in which the authors implicate SIRT1 and associated histone modifications at specific target genes in KNDy cells in the arcuate nucleus of the hypothalamus in controlling the timing of puberty and the influence of nutritional status in that timing. The authors provide several important lines of evidence to support this scheme. However, several additional lines of evidence, essential for them to make this claim, particularly for a journal like Nat Commun, are missing. The addition of the following data would greatly strengthen the

manuscript and make it suitable for the journal.

1. The authors use SIRT-Tg mice and ICV injections of a SIRT agonist (SA3) to directly implicate SIRT activity in control of Kiss1 expression in KNDy cells. This represents a good start but is not sufficient because both approaches affect SIRT activity far beyond this cell type. The authors must use some method that selectively targets KNDy cells or, at the very least, all arcuate neurons. This is very feasible by use of either Cre driver lines that target KNDy cells in conjunction with a Cre-dependent viral vector expressing SIRT1 into arcuate, or use of a non-Cre-dependent vector injected into arcuate.

2. The authors do not show whether over- or under-nutrition affects SIRT1 levels in KNDy cells specifically. They only show such regulation in dissections of hypothalamus. They show coexpression of Kiss 1 and Sirt1 but not regulation of Sirt1 expression in these cells. Showing regulation in cultured cells helps but is not good enough. The standard in the field now is to demonstrate this in vivo and the necessary tools are readily available.

3. My favorite part of the study are the data shown in Figure 6. However, these data should be improved in two main ways. First, the authors must report Sirt1, H3K9ac, H4K16ac, H3K27me3, and H3K4me3 enrichment for at least one "control" site on the Kiss1 gene. Providing such control data for the Tac3 gene would help, too, but is not necessary. A related question is how the authors selected the region of the Kiss1 and Tac3 promoters to study. (This may be in the manuscript but I didn't see it.) A more prominent explanation would be helpful.

4. As well, all data in Figure 6 should be expressed as % input. This will provide the reader with an indication of the strength of the binding. This is a very important dimension of the data, and it is now standard in the field to present qChIP data in this way.

Minor concerns:

5. The authors suggest a change in KNDy cell activity upon SIRT1 manipulations in the Results section, but this is never shown. The authors should either perform c-Fos staining or remove the word activity. They could simply say that SIRT manipulations affect KNDy cells as evidenced by altered Kiss1 expression.

6. The authors should consider citing manuscripts that implicate SIRT1 in brain in other neuropsychiatric phenomena such as addiction and depression (PMIDs: 24107942; 25698746; 27511015) if they agree that they fit, but this is not required.

Reviewer #1

“This is an extremely interesting study hypothesizing a specific epigenetic mechanism in certain hypothalamic neurons for the onset of puberty and its regulation by diet. This involves repression of the Kiss 1 gene by SIRT1 with the help of EED, leading to observed changes in histone marks at the promoter. SIRT1 transgenic mice and pharmacological SIRT1 activation are used to draw a causal relationship between SIRT1 at the Kiss1 promoter and delay of puberty. Since SIRT1 is a known nutrient sensor, the model can connect diet to onset of puberty in a satisfying way. Most of the experiments are well done and the data is convincing.”

We thank the referee for his/her positive remarks and constructive comments. Following the specific indications received, revisions have been made to the manuscript, as outlined below.

“Several modifications would strengthen the paper. First, RNA analysis for Kiss1 and Tac3 should be provided for the studies in Fig. 1 through 4. This data is the basis of the hypothesis and should be complete. For example, one expects Kiss1 RNA to change predictably in all cases.”

The referee makes a very good point. We have now carried out additional expression analyses for *Kiss1* and *Tac3* in the MBH of the animals represented in Fig.1-3. Consistent with our previous ChIP data and main conclusions of the study, we observed an inverse relation between SIRT1 content and *Kiss1* mRNA levels; these data have been included in new Figures 1-4. As predicted by our initial ChIP assays, the changes in *Tac3* expression were much less obvious (new **Suppl. Fig. S2**) stressing the importance of a predominant SIRT1/*Kiss1* regulatory pathway in the control of puberty.

“Second, in Fig. 6a the H3K4me3 does not appear to correlate well with the hypothesis. I found this confusing and would require a more detailed explanation or deletion of the experiment altogether.”

H3K4me3 is an activating histone mark catalyzed by the Trithorax group (TrxG) of proteins, which is the main epigenetic force counterbalancing the repressive actions of the PcG complex, as illustrated recently by one of our groups (Toro et al. Nat Commun 2018; 9:57). While the repressive nature of SIRT1 actions prompted us to focus our analyses on a potential putative SIRT1/PcG interplay, we considered of interest to monitor also this TrxG-dependent epigenetic modification. Our results show that early onset obesity evokes an upsurge of H3K4me3 abundance, which may contribute to accelerate the loss SIRT1/PcG repression of *Kiss1* and, hence, advance puberty onset. Because H3K4me3 abundance did not change significantly in conditions of undernutrition, alterations in TrxG influence might be more prominent in conditions of energy surplus. We have added a comment to this effect (see page 8, line 188).

“Third, the reciprocal co-IP in Fig. 7b using HA to IP should be provided.”

We agree. This is now shown in new **Fig. 8**.

“Fourth, the authors should balance their discussion with the possibility that at least some of the effects in the hypothalamic neurons may not be cell autonomous. The transgenic mice or STAC injection are systemic and may exert effects on these neurons indirectly, e.g. via adipokines. Granting that Fig. 7 in N2A cells is consistent with a cell autonomous model, only a brain specific SIRT1 transgenic mouse would rule out effects of WAT. A brain specific SIRT1 KO would also add to the study. However, I do not require additional experiments with new transgenic mice, which may take years. The authors should simply present a more sophisticated analysis of what their data actually show.”

This is a reasonable point. Anyhow, the results of our pharmacological experiments involving the central injection of the SIRT1 activator SA3, and the new experiment showing that SIRT1

overexpression in the MBH not only prevents the decrease in SIRT1 expression that antedates puberty, but also delays the pubertal process, strongly support the view that SIRT1 acts centrally to delay sexual development. These new results are shown in new **Fig. 5**, and the overall subject is discussed in page 9-10, lines 243-249; page 10, lines 260-262; and page 11, lines 284-287.

“Less important, a few key references are missing. The fact that SIRT1 is an NAD-dependent deacylase should be referenced as Imai et al., Nature 2000. The fact that SIRT1 transgenic mice show delayed puberty should be referenced by Bordone et al., Aging Cell 2007. This earlier study does not detract from the novelty of this paper, which is the epigenetic mechanism proposed.”

We apologize for inadvertently omitting these references in our initial submission. They are now included in the revised manuscript (see new references 22 and 34; and page 4, line 76 and page 6, lines 138-140).

Reviewer #2:

“Despite tremendous technical advances over recent decades, the far-reaching and apparently simple question of what initiates the onset of puberty continues to baffle endocrinologists. While this paper does not answer the question, it does identify what appears to be an important new link in the chain. Vazquez and Toro et al have shown that removal of the NAD-dependent deacetylase SIRT1 from the Kiss1 promotor modulates pubertal onset. As this conclusion seems robust, it follows logically that under- and over-nutrition effect SIRT1 levels on the Kiss1 promotor, as these nutritional states are well documented to affect kiss1 levels and puberty onset. I am enthusiastic about this paper and think that it represents a significant advancement for the field.”

We thank the referee for his positive and constructive comments, which are very much appreciated. We have now revised our paper along the lines outlined by the reviewer.

However, I have a number of questions which I think need to be addressed either here, or as discussion points.

1. *“In order to compliment the gain-of-function experiments, which importantly do not target Kiss1-neurons specifically, it would seem logical to generate mice with targeted deletion of SIRT1 in kiss1-neurons. The animals to do this are available form Jackson Labs. Did the authors attempt these experiments, or do they predict that an inducible Kiss1-cre would be required?”*

Instead of attempting to accentuate the normal loss of SIRT1 expression that occurs in the MBH preceding puberty we chose to prevent this loss via AAV-mediated delivery of SIRT1 targeted to the ARC. We felt that this approach would be more informative than attempting to generate conditional KO mice. As noted by Referee 1, generation of an inducible Kiss1-specific KO mouse would require several years of work, especially because of the current absence of mice carrying an inducible Kiss1-Cre deletion system. The results obtained using animals with SIRT1 overexpression targeted to the ARC are presented in new **Fig. 5** and described and discussed in detail in the revised text (see pages 6-7, lines 150-161; pages 9-10, lines 243-245; and page 10, lines 260-261).

2. *“The manuscript focuses on the Arcuate hypothalamic nucleus. Is there evidence that SIRT1 action in the AVPV hypothalamus is also important in puberty onset?”*

We focused our study on ARC/MBH Kiss1 neurons, as mounting evidence suggests this is the key population for activation of pulsatile GnRH neurosecretion (see Han et al. PNAS 2015; 112:13109-14; and Clarkson et al. PNAS 2017; 114(47): E10216-E10223), and for metabolic modulation of puberty onset (see Roa et al. Endocrinology 2009; 150:5016-26). Initial ChIP assays were also performed on the POA, selectively addressing the interactions of SIRT1 and the histone marks, H3K9Ac and H4K16Ac, with the *Kiss1* promoter. These assays revealed very modest modifications, if any; for this reason, analyses were not extended to the whole set of epigenetic marks evaluated in MBH samples. This information is now included as supplemental data (**Suppl. Fig. S3**), and briefly discussed in the revised manuscript (see page 8, lines 204-207; and page 10, lines 270-272).

3. *“Only approximately 50% of Kiss1 neurons express SIRT1 (Fig5). How does this fit in to the author’s model of puberty onset? Do they predict higher Kiss1-expression, and a lack of involvement in puberty-onset, in pre-pubertal SIRT1-negative cells? I think this may need a discussion point.”*

We have elaborated further this point in the revised paper. We expanded our single-cell PCR analyses to thoroughly validate the quantitative estimations of co-expression of *Sirt1* and *Kiss1* in MBH *Kiss1* neurons. In addition, we conducted immunohistofluorescence co-localization analyses to evaluate changes in SIRT1 immuno-reactivity in kisspeptin-expressing

neurons of both underfed and overfed animals. These results, presented in new **Fig. 6**, show that under-nutrition and delayed puberty were associated with increased SIRT1 content in Kiss1 neurons, whereas the opposite was observed in early-onset over-nutrition and advanced puberty. Comments along these lines have been included in the revised paper (page 7, lines 163-169; and page 10, lines 257-260).

4. *“Did the authors look at SIRT1 levels before day 10 (Fig1a)? Is it possible that the animal is born with very high SIRT1 and that it is progressively lost, at a rate that is perhaps coupled to body-weight, until it reaches a level that is permissive for Kiss1 transcription?”*

This is an interesting point, which we have addressed in our revision. We evaluated changes in hypothalamic SIRT1 content during the neonatal period (PND2 and PND6) and compared those levels with the content at PND10 (included in our initial analyses). As predicted by the referee, there is a gradual decline in total SIRT1 protein content in the hypothalamus during the neonatal-to-infantile transition, which precedes the drop occurring at later stages (juvenile and peripubertal transition). This new information might not be central to the theme of our study (which focuses on subsequent maturational events), but of interest. Accordingly, these results are displayed in a new **Suppl. Fig. S1**, and briefly presented in the Results section (see page 4, lines 99-101).

5. *“Following on from point 4, the authors may want to speculate on future efforts to determine the nature of the signal that might initiate the removal of SIRT1 from the Kiss1 promotor.”*

This is a very interesting suggestion. The signals regulating Sirt1 expression in kisspeptin neurons of the ARC are likely to be epigenetic, but they may also involve changes in transcription factor regulation taking place at both the promoter and distal enhancer regions. We have now generated RNA-seq and ChIP-seq data that we believe may help us in identifying changes in the regulation of these regulatory regions. In addition, identification of the potential extracellular signals (e.g., metabolic factors and hormones) controlling SIRT1 content and activity in kisspeptin neurons merits future investigation. While we believe these are beyond the scope of the current work, some comments regarding this point have been added to in the revised manuscript (see page 11, lines 280-287).

Reviewer #3:

“This manuscript by Vazquez, Toro and co-workers examines the role of sirtuin 1 (sirt1) in metabolic regulation of puberty. They use established models of under and over nutrition to manipulate the timing of puberty and find enhancement and repression, respectively of sirt1 in the regions of the hypothalamus containing kisspeptin neurons, which are important upstream regulators of the pubertal process. Overexpression of Sirt1 did not alter body mass but did delay puberty in a manner similar to under nutrition; pharmacological activation of sirt1 had a similar effect. Kisspeptin neurons in the arcuate were shown to express Sirt1 and ChIP was used to demonstrate nutritional regulation of the chromatin of the Kiss1 and Tac2, but not Pdyn, genes. This is an interesting set of studies that nicely complement previous work demonstrating a role for epigenetic regulation of central processes in the initiation of puberty and extending this work to physiologically-relevant models that alter the timing of puberty.”

Thank you for the positive remarks and constructive criticism.

“In figure 3, corresponding measures of Kiss1 mRNA and uterine mass (similar to figure 4) should be provided.”

We have now measured *Kiss1* mRNA levels in the MBH of this TG model, and the results are shown in a new **Fig. 3**. We also carried out additional *Kiss1* expression analyses, which are now shown in new **Fig. 1** and **Fig. 2**. The new results further support our overall hypothesis as they show a suppression of *Kiss1* expression when SIRT1 content is elevated, and *vice versa*; a feature that has been discussed in our revised manuscript (see page 10, lines 246-249). Unfortunately, we do not have measures of uterine mass from these mice because we did not collect uterine tissue at the time of sacrifice.

“In figure 5, please provide quantification of the percent of Kiss1 neurons expressing Sirt1. Likewise, given the metabolic importance on non-kisspeptin cells in the arcuate nucleus, a quantification of Kiss1 negative/Sirt1 positive cells would be informative. Both of these measures would be stronger if an effect of metabolic condition was shown to regulate coexpression.”

To address this comment, we expanded our single-cell PCR analyses and conducted immunohistochemical co-localization analyses to assess changes in SIRT1 immunoreactivity in kisspeptin-expressing neurons in our models of early over-nutrition and undernutrition. These results are presented in new **Fig. 6** and are discussed in the revised manuscript (page 7, lines 163-169; and page 10, lines 257-260). As shown in this figure, undernutrition and delayed puberty are associated with an increased SIRT1 content in *Kiss1* neurons, whereas the opposite was observed in overfed rats with advanced puberty.

“Also, in figure 5, negative controls for single-cell PCR that include RT of the solution in which the harvested cells are suspended should be included.”

Done. See new **Fig. 6**, upper panel.

The discussion nicely places the work in context, although it might be prudent to state that KNDy neurons are likely involved given the broad expression of Sirt1 in the arcuate and the lack of cell-specific modifications in the *in vivo* studies.

The referee makes a fair point. This is now clarified in the revised manuscript (see page 10, lines 261-262).

Line 246 Probably best to specify affiliation as here may be ambiguous.

We apologize for the confusion; the information has been included as suggested (see page 12, lines 307-308).

Line 293 please give final concentration of DMSO injected.

Done (see page 13, lines 350).

Line 370 R22 cells are malignant melanoma cells, what is rationale for these when cell of interest is post mitotic?

We are afraid there might be some confusion here. This cell line is actually the embryonic rat hypothalamus cell line, R22 (also termed, rHypoE-22, purchased from Cedarlane, <https://www.cedarlanelabs.com/>); see page 16. Lines 447-448. We employed this cell line because it expresses the majority of genes that are normally expressed in the developing hypothalamus.

Reviewer #4:

“This is a potentially very interesting study in which the authors implicate SIRT1 and associated histone modifications at specific target genes in KNDy cells in the arcuate nucleus of the hypothalamus in controlling the timing of puberty and the influence of nutritional status in that timing. The authors provide several important lines of evidence to support this scheme. However, several additional lines of evidence, essential for them to make this claim, particularly for a journal like Nat Commun, are missing. The addition of the following data would greatly strengthen the manuscript and make it suitable for the journal.”

Thank you. We appreciate the positive remarks and constructive comments of the referee.

1. *“The authors use SIRT-Tg mice and ICV injections of a SIRT agonist (SA3) to directly implicate SIRT activity in control of Kiss1 expression in KNDy cells. This represents a good start but is not sufficient because both approaches affect SIRT activity far beyond this cell type. The authors must use some method that selectively targets KNDy cells or, at the very least, all arcuate neurons. This is very feasible by use of either Cre driver lines that target KNDy cells in conjunction with a Cre-dependent viral vector expressing SIRT1 into arcuate or use of a non-Cre-dependent vector injected into arcuate.”*

We have now used one of the approaches suggested by the reviewer and obtained results that strongly support the central hypothesis of our paper. The results are presented in new **Fig. 5**, are described in the text (see pages 6-7, lines 150-161) and discussed on pages 9-10, lines 243-245, and page 10, lines 260-261. For a more detailed description of the rationale underlying the approach we used please our response to comment 1 from reviewer 2.

2. *“The authors do not show whether over- or under-nutrition affects SIRT1 levels in KNDy cells specifically. They only show such regulation in dissections of hypothalamus. They show coexpression of Kiss 1 and Sirt1 but not regulation of Sirt1 expression in these cells. Showing regulation in cultured cells helps but is not good enough. The standard in the field now is to demonstrate this in vivo and the necessary tools are readily available.”*

We have now carried out an immunohistofluorescence analysis to assess the effect of over- and undernutrition on the SIRT1 content in kisspeptin neurons of the ARC. As shown in a new **Fig. 6**, undernutrition and delayed puberty are associated with increased SIRT1 content in Kiss1 neurons, whereas the opposite was observed in overfed rats with advanced puberty. These new results, that strongly support the central hypothesis of our study, are discussed in the revised manuscript (page 7, lines 163-169; and page 10, lines 257-260).

3. *“My favorite part of the study are the data shown in Figure 6. However, these data should be improved in two main ways. First, the authors must report Sirt1, H3K9ac, H4K16ac, H3K27me3, and H3K4me3 enrichment for at least one “control” site on the Kiss1 gene. Providing such control data for the Tac3 gene would help, too, but is not necessary. A related question is how the authors selected the region of the Kiss1 and Tac3 promoters to study. (This may be in the manuscript, but I didn’t see it.) A more prominent explanation would be helpful.”*

This new set of data has been included in the revised paper (see new **Figure 8**). In addition, a justification of the regions of Kiss1 and Tac3 promoters selected for analyses has been included in the revised paper (see page 7, lines 173-174; and page 8, lines 192-194).

4. *“As well, all data in Figure 6 should be expressed as % input. This will provide the reader with an indication of the strength of the binding. This is a very important dimension of the data, and it is now standard in the field to present qChIP data in this way.”*

The revised qChIP figures are presented as % input.

Minor concerns:

5. *“The authors suggest a change in KNDy cell activity upon SIRT1 manipulations in the Results section, but this is never shown. The authors should either perform c-Fos staining or remove the word activity. They could simply say that SIRT manipulations affect KNDy cells as evidenced by altered Kiss1 expression.”*

The text has been modified as suggested by the reviewer; see page 7, lines 167-169.

6. The authors should consider citing manuscripts that implicate SIRT1 in brain in other neuropsychiatric phenomena such as addiction and depression (PMIDs: 24107942; 25698746; 27511015) if they agree that they fit, but this is not required.

The referee touches on an intriguing possibility. While we prefer not to be overly speculative, we also believe that this a subject of considerable interest. Therefore, we added a comment to the end of the Discussion, together with three new references (see pages 11-12, lines 296-301; and references 45, 46 and 47).

REVIEWERS' COMMENTS:

Reviewer #1 (Remarks to the Author):

This revised paper has effectively addressed all of my previous comments. It is significantly upgraded from an already strong submission. Definitely publish.

Reviewer #2 (Remarks to the Author):

The authors have provided experimental evidence for questions that I considered Discussion points. I feel that these changes have strengthened an already convincing manuscript. I am happy with the revision and have no further comments.

Reviewer #3 (Remarks to the Author):

I have carefully reviewed the author's response to my comments and those of the other reviewers and the revised manuscript. Their response has been quite comprehensive and raises the level of an already very interesting paper. This work is a considerable advance in our mechanistic understanding of the metabolic control of puberty.

In general, this paper is easy to read, but some light editing to fix a few odd verb forms and prepositions will help.

Reviewer #4 (Remarks to the Author):

The authors have done a very good job in addressing my earlier concerns. This manuscript is now suitable for publication.

Reviewer #1

“This revised paper has effectively addressed all of my previous comments. It is significantly upgraded from an already strong submission. Definitely publish.”

Response: We thank the reviewer for his/her positive comments and evaluation of our revised work.

Reviewer #2:

“The authors have provided experimental evidence for questions that I considered Discussion points. I feel that these changes have strengthened an already convincing manuscript. I am happy with the revision and have no further comments.”

Response: We appreciate the positive comments of the referee on the quality and interest of our work.

Reviewer #3:

“I have carefully reviewed the author’s response to my comments and those of the other reviewers and the revised manuscript. Their response has been quite comprehensive and raises the level of an already very interesting paper. This work is a considerable advance in our mechanistic understanding of the metabolic control of puberty. In general, this paper is easy to read, but some light editing to fix a few odd verb forms and prepositions will help”

Response: The positive comments of the referee are very much appreciated. Following his/her advise, some light editing of the manuscript has been conducted to improve English usage and readability. Examples of such editing can be found at (but are not restricted to) lines 54, 73, 86, 111, 113, 242, 254, 276, 278, 291 and 296 of the revised manuscript (labeled by the track-changes tool). We trust that these edits, together with the formal/style changes implemented as per the Editorial suggestions received, have improved the quality and readability of our paper. Further edits of the manuscript at the editorial level to meet the requirements of this referee are also welcome.

Please note that the line positions indicated above correspond to the final version of the revised manuscript, without displaying the track-changes.

Reviewer #4:

“The authors have done a very good job in addressing my earlier concerns. This manuscript is now suitable for publication.”

Response: We thank the reviewer for his/her positive evaluation of our revised work.